# JNK$^{1/2}$ represses *Lkb*$^1$-deficiency-induced lung squamous cell carcinoma progression

Jian Liu [1], Tianyuan Wang [2,11], Chad J. Creighton [3,4,5,11], San-Pin Wu [1,11], Madhumita Ray [1], Kyathanahalli S. Janardhan [6], Cynthia J. Willson [6], Sung-Nam Cho [7], Patricia D. Castro [8], Michael M. Ittmann [8], Jian-Liang Li [2], Roger J. Davis [9,10] & Francesco J. DeMayo [1]

Mechanisms of lung squamous cell carcinoma (LSCC) development are poorly understood. Here, we report that JNK1/2 activities attenuate *Lkb1*-deficiency-driven LSCC initiation and progression through repressing ΔNp63 signaling. In vivo *Lkb1* ablation alone is sufficient to induce LSCC development by reducing MKK7 levels and JNK1/2 activities, independent of the AMPKα and mTOR pathways. JNK1/2 activities is positively regulated by MKK7 during LSCC development. Pharmaceutically elevated JNK1/2 activities abates *Lkb1* dependent LSCC formation while compound mutations of *Jnk1/2* and *Lkb1* further accelerate LSCC progression. JNK1/2 is inactivated in a substantial proportion of human LSCC and JNK1/2 activities positively correlates with survival rates of lung, cervical and head and neck squamous cell carcinoma patients. These findings not only determine a suppressive role of the stress response regulators JNK1/2 on LSCC development by acting downstream of the key LSCC suppresser *Lkb1*, but also demonstrate activating JNK1/2 activities as a therapeutic approach against LSCC.

[1] Reproductive & Developmental Biology Laboratory, National Institute of Environmental Health Sciences (NIEHS), Research Triangle Park (RTP) 27709 NC, USA. [2] Integrative Bioinformatics, National Institute of Environmental Health Sciences (NIEHS), Research Triangle Park (RTP) 27709 NC, USA. [3] The Dan L. Duncan Cancer Center, Baylor College of Medicine (BCM), Houston 77030 TX, USA. [4] Department of Medicine, Baylor College of Medicine (BCM), Houston 77030 TX, USA. [5] Department of Bioinformatics and Computational Biology, The University of Texas MD Anderson Cancer Center, Houston 77030 TX, USA. [6] Integrated Laboratory Systems, Research Triangle Park 27709 NC, USA. [7] Department of Molecular and Cellular Biology, BCM, Houston 77030 TX, USA. [8] Department of Pathology & Immunology, BCM, Houston 77030 TX, USA. [9] Howard Hughes Medical Institute (HHMI), University of Massachusetts Medical School, Worcester 01655 MA, USA. [10] Program in Molecular Medicine, University of Massachusetts Medical School, Worcester 01655 MA, USA. [11] These authors contributed equally: Tianyuan Wang, Chad J. Creighton, San-Pin Wu. Correspondence and requests for materials should be addressed to F.J.D. (email: francesco.demayo@nih.gov)

Precision medicine has been enjoying successes in recent years, especially with the application of genomics-driven cancer therapy[1]. For example, the discovery of key drivers (e.g. *EGFR* and *ALK*) of human lung adenocarcinoma using genetic mouse models has led to the development of targeted therapies[2]. Therefore, identifying cancer driver mutations in human cancers is valuable for targeted therapy[2,3].

Lung squamous cell carcinomas (LSCCs) comprise about 25–30% of all lung cancers, which is the leading cause of cancer-related death in the United States and worldwide[4]. Human LSCC develops from normal airway epithelium and progresses through hyperplasia, squamous metaplasia, dysplasia, and carcinoma[5]. SCC characteristics emerge at the squamous metaplasia stage in the nests of neoplastic squamous cells with an increase in marker expression (e.g. p63)[6]. Genomic analysis of human LSCC tumors has provided a comprehensive landscape of the molecular alterations, including frequently mutated genes (e.g., *TP53*, *PTEN* and *LKB1*) and altered pathways (e.g., SOX2, PI(3)K/LKB1, EGFR/FGFR/RAS) associated with this disease[7,8]. Several pre-clinical models show that mice carrying compound mutations develop LSCC, as evidenced by studies on *Kras*[G12D]/*Lkb1*[loss], *Sox2*[OX]/*Lkb1*[loss], *Pten*[loss]/*Lkb1*[loss], and *Cdkn2a*[loss]/*Pten*[loss]/*Sox2*[OX] mice[8–11]. In contrast, mutations in one single gene lead to lung adenocarcinomas (ADs), as shown by *Trp53* deficiency, *Pten* ablation, *Kras*[G12D] or *Egfr*[L858R] activation and *Sox2* over-expression (*Sox2*[OX])[10,12–14]. To date, the question on whether LSCC can be initiated by a single gene mutation remains outstanding and the underlying mechanism of LSCC formation and progression remains unclear[4,15].

In addition to genetic causes, LSCC development is frequently impacted by environmental stresses[16], as documented in exposures to smoking[17], air pollution[18] and radon[19]. Upon environmental challenges, stress signaling, such as the MKK7-JNK1/2 pathway, participates in induction of inflammation responses and changes of expression of cancer-related genes[20–22]. The JNK1/2 signaling has been linked to cancers in lung, breast and colon and ovary, mainly related to adenocarcinoma cells[23]. However, the JNK1/2 pathway could act as either a positive or a negative regulator in tumorigenesis[24–26]. It is not clear what is the role of JNK1/2 signaling in SCC development despite that JNK1/2 is related to LSCC-associated environmental challenges.

*LKB1* (also called *STK11*, serine/threonine kinase 11) is a frequently mutated gene in human LSCCs[8,9]. *LKB1* has been shown as an important suppressor of LSCC, supported by observations that *Lkb1* deficiency in mouse lungs drives LSCC development in conjunction with additional mutations[8–10]. However, ablation of *Lkb1* alone in mouse lung using adenovirus-Cre (Ad-Cre) via intranasal delivery does not cause pulmonary neoplasia[8,9]. This may be due to the limitation on efficiency of intranasal delivery of Cre into large airways where human LSCC is frequently initiated[5,27]. Using a previously generated codon-optimized Cre recombinase under the control of the Club Cell Secretory Protein promoter (*CCSP*[iCre]) for all airway epithelial cells, including cells of large airways, we found that *Lkb1* deficiency by itself is sufficient to induce LSCC. In contrast, no LSCC was induced after manipulation of other five candidate genes that have frequent mutations in human LSCCs, including *TP53*, *PTEN*, *ERRFI1*, *SMAD4* and *KRAS*[7]. We further identified the stress response pathway MKK7-JNK1/2 as downstream effectors of *Lkb1* in suppressing ΔNp63 signaling during LSCC development. The observations that LSCC patients with higher JNK1/2 activities have better survival rates and activating JNK1/2 activities attenuate LSCC progression suggest targeting the JNK1/2-mediated stress response pathway as a way to combat against LSCC.

## Results

**Lkb1 deficiency alone is sufficient to induce LSCC.** Ablation of *Lkb1* in mouse airway was achieved by crossing *Lkb1* floxed mice with *CCSP*[iCre] mice (*Lkb1*[d/d]: *CCSP*[iCre]*Lkb1*[f/f]) (Supplementary Fig. 1a–b). Strikingly, LSCCs were found in *Lkb1*[d/d] mice, as early as, 11 months old in addition to AD (Fig. 1a and Supplementary Fig. 2a–b; Table 1 and Supplementary Data 1−2). LSCC lesions in *Lkb1*[d/d] mice displayed typical nests of neoplastic squamous cells, some with keratinization and most with infiltrates of neutrophils (Fig. 1a). Cells in the lesions exhibited positive staining of LSCC clinical markers (p63, ΔNp63 and CK5) and lack of lung AD marker TTF1 expression (Fig. 1b and Supplementary Fig. 1c–d)[28]. Two subpopulations of SCC cells are observed at lesions sites based on levels of LSCC markers p63 and ΔNp63. Cells in nests of neoplastic squamous cells (Fig. 1b, region IV and Supplementary Fig. 1c–d) exhibited higher levels of both p63 and ΔNp63. In contrast, SCC cells adjacent to large airways (Fig. 1b, region III and Supplementary Fig. 1c–d) have relatively lower expression of p63 and ΔNp63, which could be cells at the early phase of LSCC. Meanwhile, progressive lung SCC developmental stages (SCC-DSs) were also observed after *Lkb1* ablation (Supplementary Data 1-2), including epithelial hyperplasia (5.4%) (Supplementary Fig. 1e), squamous metaplasia (1.8%) (Supplementary Fig. 1e), adenocarcinoma with squamous differentiation (10.7%) (Supplementary Fig. 1f) and adenosquamous carcinoma (ASC) (5.4%) (Supplementary Fig. 1g). These lung SCC-DSs showed the focal or diffuse positive staining of p63 and CK5, respectively, as well as the focal weak or negative staining of TTF1 (Supplementary Fig. 1e–g). On the other hand, unlike *Lkb1* ablation, no lung SCCs and SCC-DSs were induced after individual manipulation of five known SCC-associated genes *Pten*, *p53*, *Errfi1* (*Mig6*), *Kras* and *Smad4* (Table 1 and Supplementary Data 1−2). Instead, lungs of *Pten* deletion, *P53* ablation, *Errfi1* knockout or *Kras*[G12D] activation had AD tumors, adenomas and atypical adenomatous hyperplasia that showed opposite profiles of marker expression compared with lung SCCs in *Lkb1*[d/d] mice (Table 1 and Supplementary Fig. 2b–f)[8,29]. Moreover, loss of *Smad4* exhibited no histomorphologic change in lungs (Supplementary Fig. 2g–h), consistent with a previous report[13]. In summary, these findings reveal a unique capacity of *Lkb1* deficiency in the induction of lung SCCs and SCC-DSs.

Transcriptome profiling demonstrates that the *Lkb1*-deficient mouse LSCC exhibited a high degree of positive correlation with human LSCC, especially with LSCC Subtypes 2a and 2b, compared to other subtypes of human lung cancers (Fig. 1c and Supplementary Fig. 3a)[30]. These findings support that *Lkb1*[d/d] mouse LSCCs closely resemble human LSCCs at the transcriptome level. Functionally, *Lkb1* deficiency altered expression of genes that are enriched in functions of cell movement, morphology, death and survival (Supplementary Table 1), reflecting the histological observations. Interestingly, the known LKB1 downstream target AMPKα[31] did not change its phosphorylation levels in the absence of LKB1 at the SCC stage (Fig. 1d). Another LKB1 target mTOR even showed reduced levels of phosphorylation, in contrary to the previously reported role of LKB1 in suppressing mTOR activity (Fig. 1d)[32]. Like the observations in SCCs, no significant changes of AMPKα and mTOR phosphorylation levels were found in 1- and 3-month old mouse lungs at the pretumor stage in response to *Lkb1* deficiency (Fig. 1e and Supplementary Fig. 3b–c). These results suggest that LKB1 utilizes pathways other than those of known cancer-associated ones for LSCC development. Close examination of genes altered by *Lkb1* deficiency via ingenuity pathway analysis (IPA) revealed JNK1/2, P38, NFκB and ERK1/2 as the common

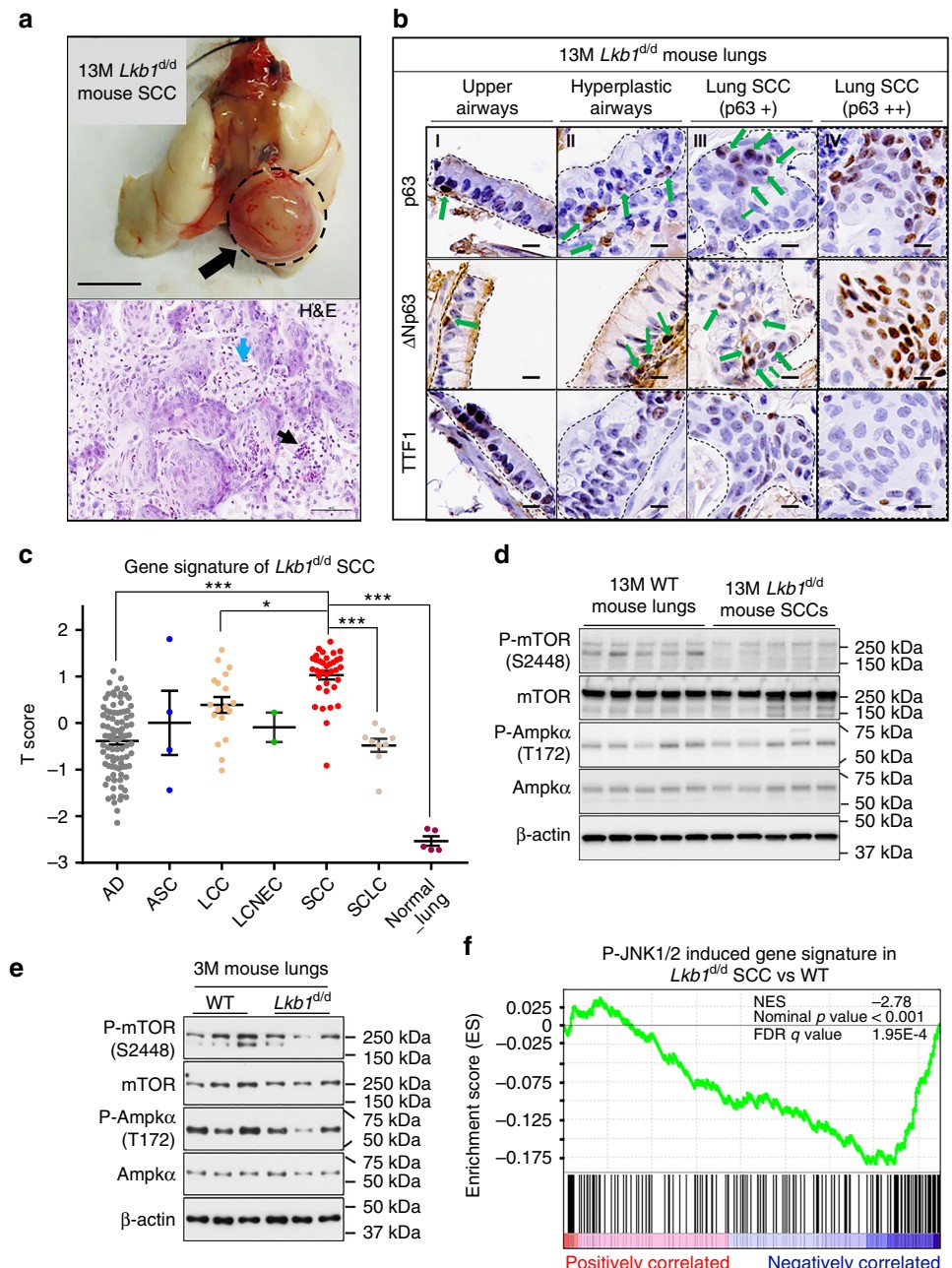

**Fig. 1** *Lkb1* ablation alone induces LSCC development with negative JNK1/2 activities. **a** Upper panel: Gross appearance of lung tumors in 13-month-old (13M) *CCSP*[piCre]*Lkb1*[f/f] (*Lkb1*[d/d]) mice. SCC squamous cell carcinoma; SCC was labeled by a dash cycle and an arrow (upper panel). Scale bar: 0.5 cm. Lower panel: Representative 13M *Lkb1*[d/d] mouse lung SCC (H&E). Infiltrates of neutrophils were commonly associated with SCC nests (black arrow, lower panel). Keratinization was frequently present in SCCs (blue arrow, lower panel). Scale bar: 60 μm. **b** Representative immunohistochemistry (IHC) staining of 13M *Lkb1*[d/d] mouse lung SCC. SCC markers: p63 and ΔNp63; AD marker: TTF1. "+" means the expression is mild or moderate, and "++" means the expression is strong. Dashes outline: Regions of interest. Positive staining is indicated in some panels by green arrows. Scale bar: 10 μm. **c** Alignment of the gene signature of 13-month-old *Lkb1*[d/d] mouse lung SCC tumors with Takeuchi human lung cancer database (GSE11969). A score greater than zero indicates a positive correlation with the murine gene signature of 13-month-old *Lkb1*[d/d] mouse. AD adenocarcinoma, ASC adenosquamous carcinoma, LCC large-cell carcinoma, LCNEC large-cell neuroendocrine carcinoma, SCC squamous cell carcinoma, SCLC small cell lung cancer. ANOVA (Tukey−Kramer Multiple Comparisons (TKMC) Test) was performed and error bar is SEM; *$P < 0.05$; ***$P < 0.001$. **d** Western blot (WB) analysis ($n = 5$) of protein expression in 13-month-old (13M) mice. **e** WB analysis ($n = 3$) of protein expression in 3-month-old (3M) mice. **f** Gene set enrichment analysis (GSEA) of pre-ranked genes detected in microarrays analysis of 13-month-old *Lkb1*[d/d] mouse lung SCCs. All detected genes (Supplementary Data 3) were ranked based on their fold changes (*Lkb1*[d/d] SCC vs. WT) from high (the left terminal of *X*-axis) to low (the right terminal of *X*-axis)

downstream pathways for candidate upstream regulators of *Lkb1* responsive genes (Table 2 and Supplementary Data 3). Gene set enrichment analysis (GSEA) further showed that the P-JNK1/2 induced pathway, as the top enriched pathway, negatively correlated with the *Lkb1* deficiency gene signature (Fig. 1f and Supplementary Table 4). These observations establish an association between *Lkb1* deficiency and reduced JNK1/2 activities in LSCC.

### Table 1 Summary of mouse phenotypes of lung tumors and lungs

| Genotype | No. | Age (Month) | The percent (%) of phenotypes in mouse lungs | | | | | |
|---|---|---|---|---|---|---|---|---|
| | | | SCC-DS | SCC | SCC-DS + SCC | AD-DS | AD | AD-DS + AD |
| $Lkb1^{d/d}$ | 56 | 11–14 | 25.0 | 16.1 | 32.8 | 16.1 | 37.5 | 53.6 |
| $Pten^{d/d}$ | 12 | 12–13 | 0 | 0 | 0 | 16.7 | 8.3 | 25.0 |
| $P53^{d/d}$ | 10 | 12–13 | 0 | 0 | 0 | 10.0 | 10.0 | 20.0 |
| $Mig6^{d/d}$ | 12 | 9 | 0 | 0 | 0 | 8.3 | 0 | 8.3 |
| $Kras^{G12D}$ | 8 | 1 | 0 | 0 | 0 | 100 | 37.5 | 100 |
| $Smad4^{d/d}$ | 9 | 11–13 | 0 | 0 | 0 | 0 | 0 | 0 |
| WT | 25 | 1–14 | 0 | 0 | 0 | 4.0 | 0 | 4.0 |

$Lkb1^{d/d}$: $CCSP^{iCre}$ $Lkb1^{f/f}$; $Pten^{d/d}$: $CCSP^{iCre}Pten^{f/f}$; $P53^{d/d}$: $CCSP^{iCre}p53^{f/f}$; $Mig6^{d/d}$: $CCSP^{iCre}Mig6^{f/f}$; $Kras^{G12D}$: $CCSP^{iCre}Kras^{G12D}$; $Smad4^{d/d}$: $CCSP^{iCre}Smad4^{f/f}$; WT wild type. SCC development stages include epithelial hyperplasia, epithelial hyperplasia with squamous metaplasia, adenocarcinoma with squamous differentiation and adenosquamous carcinoma
Adenocarcinoma development stages include epithelial hyperplasia and adenoma
SCC squamous cell carcinoma, DS development stage, AD adenocarcinoma, No. number

### Table 2 Analysis of enriched upstream regulators and its core downstream pathways

| Top upstream regulators | P value of overlap | Core downstream pathways of TNF and IL1B |
|---|---|---|
| TGFB1 | 4.32E-41 | JNK1/2 pathway |
| TNF | 7.04E-39 | p38 pathway |
| TP53 | 2.13E-28 | NFκB pathway |
| ERBB2 | 3.54E-26 | ERK1/2 pathway |
| IL1B | 7.85E-22 | |

Top enriched upstream regulators were identified from these differentially expressed genes identified in the microarray (13-month-old (13M) $Lkb1^{d/d}$ mouse lung SCCs vs. 13M $Lkb1^{f/f}$ mouse lungs; Supplementary Data 3) using ingenuity pathway analysis (IPA)
SCC squamous cell carcinoma

**JNK1/2 is inactivated in *Lkb1*-null induced mouse LSCCs.** To further investigate the association between reduced JNK1/2 activities and LSCC development, we examined the phosphorylation of JNK1/2, P38, NFκB and ERK1/2. Indeed, JNK1/2 phosphorylation was almost abolished in $Lkb1^{d/d}$ mouse LSCCs with the most robust change among all tested candidates (Fig. 2a). Reduced JNK1/2 phosphorylation is also seen in LSCC compared to normal upper airway tissues by immunohistochemistry (IHC) assay (Fig. 2b), consistent with western blot results. Moreover, JNK1/2 phosphorylation has an opposite expression pattern to that of ΔNp63 and p63 (Fig. 1b), which further supports a negative correlation between JNK1/2 activities and LSCC in situ.

Next, the timing of change of JNK1/2 activities in response to *Lkb1* deficiency was examined to determine the sequence of events between JNK1/2 inactivation and LSCC development. We took advantage of the $CCSP^{iCre}Lkb1^{f/f}Pten^{f/f}$ ($Lkb1^{d/d}Pten^{d/d}$) mouse model that has accelerated LSCC development in 3 months (Supplementary Fig. 3d–e; Supplementary Table 2; Supplementary Data 1–2 and 4), an incidence rate of lung adenosquamous cell carcinomas (ASCs) at 88.9% (Supplementary Table 2 and Supplementary Data 2), nearly identical molecular profiles between lung SCC and ASC (Supplementary Fig. 3d−g), close resemblance of the combined SCC/ASC transcriptome profile to that of human lung SCCs (Supplementary Fig. 3h and Supplementary Data 3) and similar enriched upstream regulators (Supplementary Table 3). In addition, epithelial hyperplasia (11.1%) and epithelial hyperplasia with squamous metaplasia (11.1%) were observed in the $Lkb1^{d/d}Pten^{d/d}$ mouse model (Supplementary Data 1-2). Most importantly, JNK1/2 activities were reduced in SCCs/ASCs (Fig. 2c, d and Supplementary Fig. 4a). The pretumor stage of $Lkb1^{d/d}Pten^{d/d}$ mice was defined at 1 month old of age because at this time only hyperplasia was found in the large airways (Supplementary Fig. 5a). A kinome

array assay revealed that reduced phosphorylation levels of JNK1/2, ERK1/2 and p38α as well as increased AKT phosphorylation occur in $Lkb1^{d/d}Pten^{d/d}$ lungs at the pretumor stage (Fig. 2e; Table 3; Supplementary Fig. 4b–c and Supplementary Table 5). The western blot assay validated that only P-JNK1/2 is consistently downregulated in both SCCs and ASCs and in the pretumor stages (Fig. 2d and Supplementary Fig. 4c). The reduced JNK1/2 phosphorylation in $Lkb1^{d/d}Pten^{d/d}$ mouse hyperplastic epithelial cells was confirmed by the IHC assay (Fig. 2f). Histologically normal *Lkb1*-deficient airways have uniform p-JNK1/2, whereas hyperplastic lesions start to lose p-JNK1/2 positivity (Fig. 2b, f). This suggests that the impact of *Lkb1* deletion on p-JNK1/2 levels does not appear to be entirely direct, implying that proliferation induced by *Lkb1* loss occurs prior to the decline in p-JNK1/2 levels. The declined P-JNK1/2 levels may contribute to the development of hyperplastic lesions. As a feedback regulation mechanism, these hyperplastic lesions may facilitate the further decline of P-JNK1/2 by providing an oncogenic environment. These findings show that JNK1/2 inactivation occurs before tumor formation.

**Decreased MKK7 reduced JNK1/2 phosphorylation in mouse LSCC.** To investigate the underlying mechanism of JNK1/2 inactivation during LSCC development, we examined expression levels of MKK7, which phosphorylates JNK1/2 at T183/Y185 to regulate JNK1/2 activities in previous reports[20,33,34]. Both mRNA and protein expressions of MKK7 were indeed decreased in $Lkb1^{d/d}$ mouse LSCCs (Fig. 3a). The decrease of MKK7 expression occurs at the pretumor stage (Fig. 3b, c), sharing a similar pattern with that of the decreasing JNK1/2 phosphorylation along the course of LSCC development (Fig. 2c, f). Functionally, MKK7's impact on JNK1/2 phosphorylation in LSCC development was assessed on a mouse LSCC cell line (mLSCC) that was generated from LSCC tumors of $Lkb1^{d/d}Pten^{d/d}$ mice. Exogenous expression of MKK7 in mLSCC by transfection restored JNK1/2 phosphorylation and decreased the levels of p63 and ΔNp63 (Fig. 3d, e). Similar results were observed in H157 cells, which are human LSCC cells with the mutations of *LKB1* and *PTEN* (Supplementary Fig. 5c). Meanwhile, at the pretumor stage modeled by a nontumorigenic human bronchial epithelial cell line NL20 and Beas-2B, the *MKK7* knockout also had reduced JNK1/2 phosphorylation levels (Fig. 3g and Supplementary Fig. 5e) while *LKB1* ablation decreased levels of both MKK7 expression and phosphorylated JNK1/2 (Fig. 3f and Supplementary Fig. 5d). These results suggest that decreased MKK7 expression has a negative impact on JNK1/2 phosphorylation levels during LSCC development. Furthermore, ablation of *LKB1* and *MKK7* in NL20 and Beas-2b promoted cell growth (Fig. 3h and Supplementary Fig. 5f). These results not only delineate the

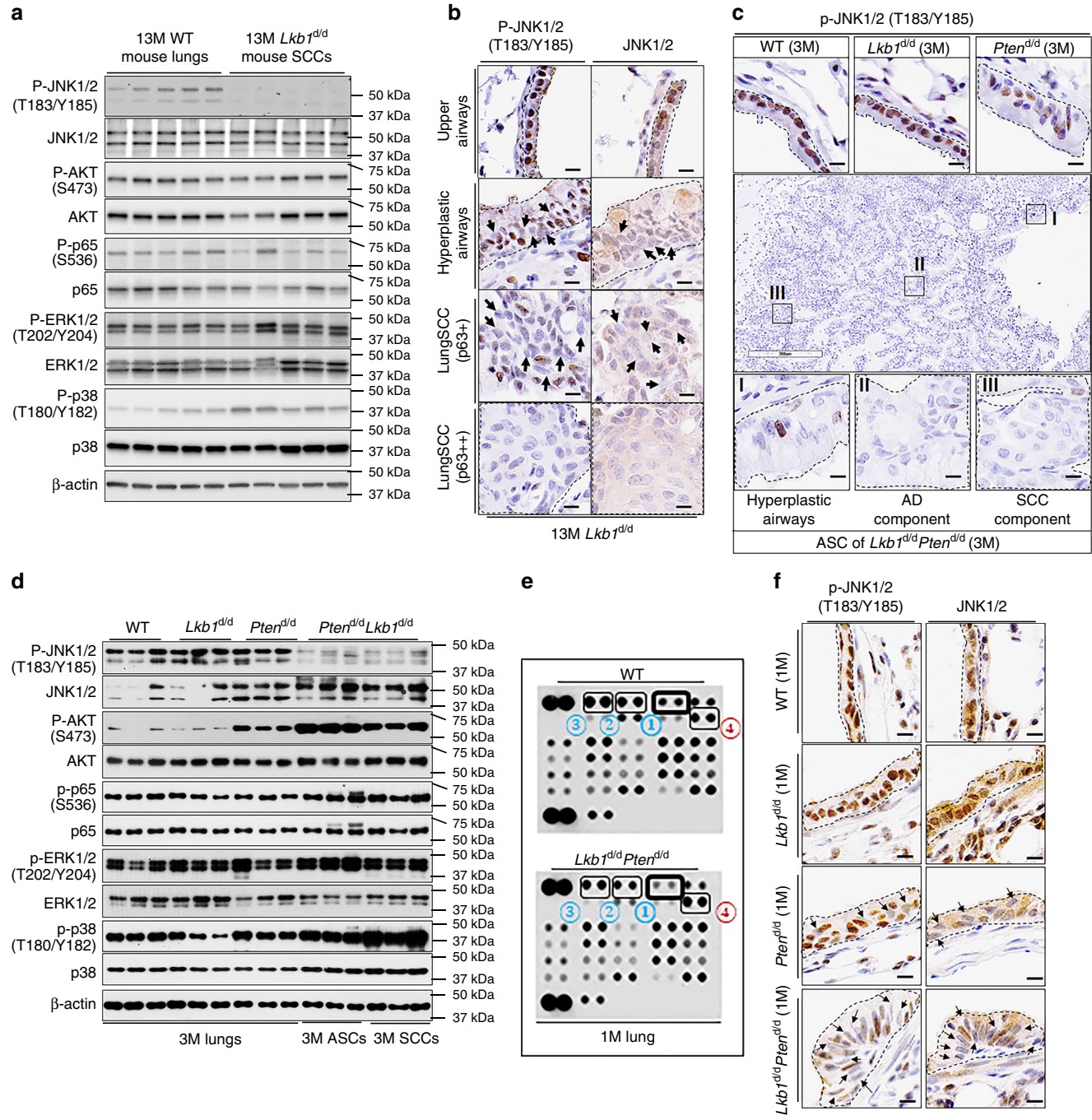

**Fig. 2** p-JNK1/2 are reduced in *Lkb1*-deficiency-driven mouse LSCC development. **a** Western blot (WB) analysis ($n = 5$) of protein expression in 13-month-old (13M) *Lkb1*d/d mice. **b** Representative P-JNK1/2 and JNK1/2 IHC staining of mouse lungs ($n = 6$). Dashes outline regions of interest. P-JNK1/2 staining is usually nuclear while JNK1/2 staining is usually cytoplasmic and nuclear. **c** Representative IHC staining results of P-JNK1/2 in 3-month-old mouse lungs and lung tumors ($n = 6$). Black dashes point to the areas of interest. Box I: hyperplastic airway; Box II: adenocarcinoma (AD) component; Box III: peripheral squamous cell carcinoma (SCC) component. Scale bar: 300 µm (black box) and 10 µm (dark bar). **d** WB analysis of 3-month-old (3M) mouse lungs and lung tumors. **e** Kinome array was performed in 1-month-old (1M) mouse lungs ($n = 6$) (see Supplementary Fig. 4b for a full summary of kinome array). The dots in the membranes are anti-phosphorylated (P)-proteins and the numbers pointing out the top changed proteins were listed in Table 3. **f** Representative IHC staining of P-JNK1/2 and JNK1/2 in 1-month-old mouse lungs ($n = 5$). Arrows point to the cells with weak nuclear staining. Scale bar: 300 µm (Black box) and 10 µm (dark bar)

LKB1-MKK7-JNK1/2 pathway at the early phase of LSCC development, but also establish a regulatory function of LKB1 on the MKK7-JNK1/2 signaling that is regarded as a stress response pathway in other systems.

**Jnk1/2 loss accelerates Lkb1-null induced LSCC development.** Since the *Lkb1*-deficient LSCC has a suppressed MKK7/JNK1/2 pathway and JNK1/2 have been indicated in tumor formation[24,34,35], we hypothesize that JNK1/2 plays a role in the suppression of LSCC development. To test this, in vitro ablation of *Jnk1/2* was first conducted in mLSCC cells to investigate its effect on cellular activities. As anticipated, *Jnk1/2* ablation increased cell growth in mLSCC cells (Fig. 4a and Supplementary Fig. 6a–b). The in vivo role of JNK1/2 in LSCC development

| Table 3 Top changed phosphorylated (P)-proteins in *Lkb1*d/ dPten^(d/d) lungs (pretumor stage) compared to wild-type (WT) lungs at 1 month of age | | |
|---|---|---|
| | **Top changed phosphorylated protein** | **Change direction** |
| ① | JNK pan (T183/Y185, T221/Y223) | Down |
| ② | Erk1/2 (T202/Y204, T185/Y187) | Down |
| ③ | P38α (T180/Y182) | Down |
| ④ | AKT (S473) | Up |

was next investigated by generating mice with the conditional ablation of both *Lkb1* and *Jnk1/2* (*Lkb1*^(d/d)*Jnk1*^(d/d)*Jnk2*^(−/−)) (Supplementary Fig. 6c). *Lkb1*^(d/d)*Jnk1*^(d/d)*Jnk2*^(−/−) mice started to develop squamous metaplasia in the lungs at 4 months old, followed by full penetrance of pathologic phenotypes (CK5+, P63+) at 7 months old and beyond (Fig. 4b and Supplementary Fig. 6d). The associated phenotypes include epithelial hyperplasia (66.7%) (Fig. 4c), epithelial hyperplasia with squamous metaplasia (25%) (Fig. 4b), and adenocarcinoma with squamous differentiation (16.7%) (Supplementary Fig. 6e), ASC (25%) (Fig. 4d) and SCC (33.3%) (Supplementary Fig. 6d) (Supplementary Data 1−2). Compared to 11–14-month-old *Lkb1*^(d/d) mice that showed epithelial hyperplasia (5.4%) (Supplementary Fig. 1c–d), epithelial hyperplasia with squamous metaplasia (1.8%) (Supplementary Fig. 1e), squamous metaplasia (1.8%), adenocarcinoma with squamous differentiation (10.7%) (Supplementary Fig. 1f), ASC (5.4%) (Supplementary Fig. 1g), SCC (16.1%) (Fig. 1a, b) (Supplementary Data 1−2), and loss of *Jnk1/2* in the *Lkb1*-deficient background accelerated LSCC development in vivo. *Lkb1*^(d/d)*Jnk1*^(d/d)*Jnk2*^(−/−) mice also developed adenocarcinoma (Supplementary Fig. 6f). Although the ADs generally showed a papillary morphology microscopically, they resembled LSCCs in that they are positive for IHC expression of the SCC markers p63 and CK5 (Supplementary Fig. 6f). Interestingly, there is no LSCC formation in *Jnk1*^(d/d)*Jnk2*^(−/−) mice, which only has adenoma/adenocarcinoma development (33.3%) (Supplementary Fig. 6g and Supplementary Data 1), concluding that the JNK1/2 pathway regulates LSCC development under a *Lkb1*-deficient background. These results suggest that *Jnk1/2* knockout promotes *Lkb1*-deficiency-induced SCC formation by inducing SCC marker expression, such as p63. Taken together, these results reveal that compound mutations of *Jnk1/2* and *Lkb1* facilitate LSCC development and demonstrate that JNK1/2 act as major suppressors for *Lkb1*-dependent LSCC progression.

**JNK1/2 inactivation activates mouse LSCC ΔNp63/p63 pathway.** We next investigated the downstream effectors of JNK1/2 in regulation of lung SCC development. ΔNp63 is a squamous cell lineage marker promoting squamous cell stratification and tumor progression[36,37]. We had shown that ΔNp63 was induced in hyperplastic airways and lung SCCs (Fig. 1b and Supplementary Fig. 3f) and was not detected in mouse lung ADs (Supplementary Fig. 1h and 2b). Based on the observation that expression patterns were opposite between levels of ΔNp63 expression (Fig. 1b and Supplementary Fig. 3f) and JNK1/2 phosphorylation (Fig. 2b, f) during lung SCC development, we examined the effect of *Jnk1/2* on the expression of ΔNp63. *Jnk1/2* knockout mLSCC cells exhibited an increase in ΔNp63 levels (Fig. 4e). Conversely, treatment with stimuli of JNK1/2 pathway, IL-1β, TNFα or Anisomycin raised JNK1/2 phosphorylation levels and decreased the expression of ΔNp63 and p63 in parental or gRNA-Control mLSCC cells (Fig. 4e)[38]. Quantification of the protein bands of ΔNp63 and p63 showed JNK1/2 had a stronger effect in repressing ΔNp63 than inhibiting p63 expression (Fig. 4e and

Supplementary Table 6). Global assessment of the impact of *Jnk1/2* ablation on gene expression revealed increased *P63* mRNA levels, altered expression of downstream targets of TP63, and changes in genes that are enriched for regulating cancer and cell proliferation (Table 4; Supplementary Table 7 and Supplementary Data 3). Downstream targets of TNF and IL-1β are also impacted by *Jnk1/2* ablation (Table 4 and Supplementary Data 3), which is in accordance with the well-known role of JNK1/2 in mediating activities of these two cytokines[38–40]. In summary, these results indicate that JNK1/2 inactivation activates the ΔNp63/p63 pathway to promote LSCC development.

**JNK1/2 activation attenuates mouse LSCC development.** While JNK1/2 activities were reduced in LSCC, JNK1/2 proteins remain present in both tumor and pretumor stages of cells (Fig. 2a, b, d, f and Supplementary Fig. 4c). This observation raises the question whether manipulating JNK1/2 activities may have an impact on *Lkb1*-dependent tumor formation. JNK1/2 activator Anisomycin-treated mLSCC cells showed a significant increase of JNK1/2 phosphorylation levels and the expression of apoptosis markers, cleaved PARP and Caspase-3, whereas the JNK1/2 inhibitor SP600125 reduced levels of these apoptotic indicators (Fig. 5a)[38,41]. Less than 20% of mLSCC cells survived after Anisomycin treatment, in contrast to the *Jnk1/2* knockout significantly maintaining cell survival (Fig. 5b, c). Consistent with the in vitro results, treatment of *Lkb1*^(d/d)*Pten*^(d/d) mice with Anisomycin for 8 weeks significantly inhibited LSCC formation (Fig. 5d, e) and lung tumor development (Supplementary Fig. 7), with no obvious inhibition on lung adenoma/adenocarcinoma (Supplementary Data 1). In contrast, Anisomycin did not affect the development of LSCC and lung tumors in *Lkb1*^(d/d)*Jnk1*^(d/d)*Jnk2*^(−/−) mice (Fig. 5f and Supplementary Data 1). These findings demonstrate that raising JNK1/2 activities can inhibit LSCC cell growth in vitro and tumor progression in vivo, suggesting that targeting JNK1/2 may benefit LSCC patients who have low JNK1/2 activities.

**JNK1/2 is inactivated in a large proportion of human LSCCs.** To further investigate the clinical relevance of JNK activities, JNK1/2 phosphorylation levels were surveyed in the TCGA protein array database[30,42] and assayed using IHC in human lung cancer tissue arrays. JNK1/2 phosphorylation levels in a sizable proportion of human LSCCs was lower than the average level in the TCGA protein array database, especially when compared to human lung AD cases (Fig. 6a). Significantly lower levels of JNK1/2 phosphorylation in SCCs compared to normal airways were also observed in two independent sets of human LSCC tissue arrays (Fig. 6b–d and Supplementary Fig. 8a–b). In a subset of patients, hyperplastic airway cells also exhibited lower levels of JNK1/2 phosphorylation compared with normal airways (Fig. 6b and Supplementary Fig. 8a–b), resembling the mouse phenotype (Fig. 2b, c, f). In summary, JNK1/2 inactivation is present in a substantial proportion of human LSCCs.

**JNK1/2 activities predict survival rates of SCC patients.** Gene signature analysis of LSCC patients was then investigated to determine whether JNK1/2 activities could predict patient survival. JNK1/2 activities were defined by a gene signature that was generated using these DEGs induced by knocking out *Jnk1/2* in mLSCC cells (Supplementary Data 3). Each TCGA human LSCC patient's JNK1/2 activities were estimated by applying mouse *Jnk1/2* knockout gene signatures to expression profiles of human tumors[43]. Survival analysis showed that patients with higher JNK1/2 activities in LSCC tumors had a better survival rate and longer relapse-free survival (RFS) than those with lower JNK1/2 activities (Fig. 6e and Supplementary Fig. 8c-–d). In contrast,

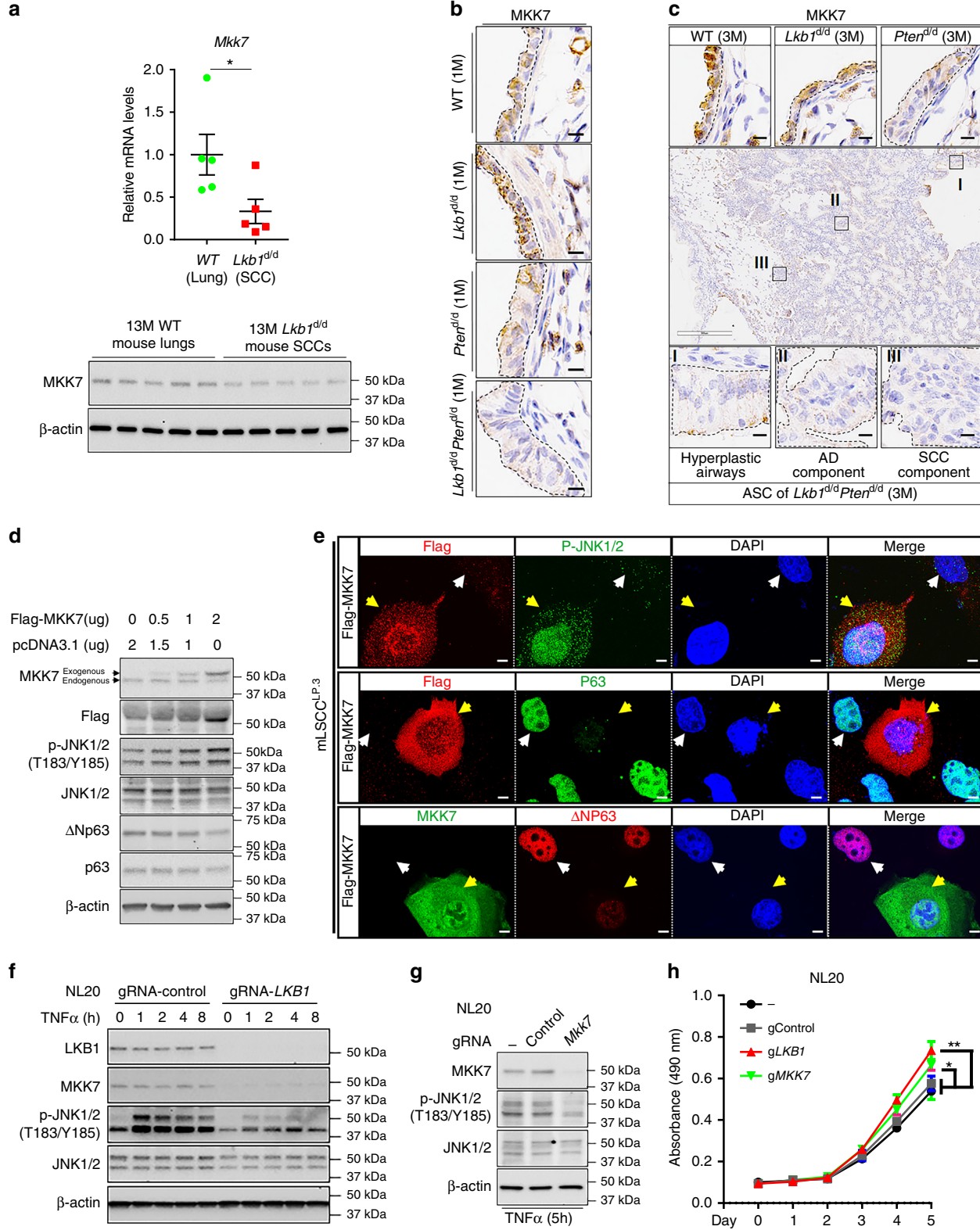

JNK1/2 activities were unable to predict the survival of lung adenocarcinoma patients (Supplementary Fig. 8e–f). In conjunction with the findings on JNK1/2 activities' antitumor function in mice (Fig. 5), these results suggest a beneficial effect of increasing JNK1/2 signaling for treatment of human LSCCs.

Considering that JNK1/2 activation is required to induce cell apoptosis in multiple SCC cells[44–47], we generated the tissue-specific JNK1/2 gene signature to predict the survival of other human SCC patients. Notably, survival analysis showed that the patients whose cervical or head and neck SCC tumors had higher JNK1/2 activities had better survival rate than those with lower JNK1/2 activities (Fig. 6f, g). These results suggest that the JNK1/2 pathway is a conserved suppressor for SCC in multiple tissues.

**Fig. 3** Decreased MKK7 in response to *Lkb1* deficiency reduces JNK1/2 activities. **a** qRT-PCR (upper panel) and WB (lower panel) analyses of the mRNA and protein expression of MKK7 in mouse lungs and lung SCCs, respectively ($n = 5$). Error bar is SEM; T test: *$P < 0.05$. **b** Representative IHC staining of MKK7 in 1-month-old mouse lungs ($n = 5$). Black dashes point to the areas of interest. Scale bar: 10 μm. **c** Representative IHC staining results of MKK7 in 3-month-old mouse lungs and lung tumors ($n = 6$). Black dashes point to the areas of interest. Box I: hyperplastic airway; Box II: adenocarcinoma (AD) component; Box III: peripheral squamous cell carcinoma (SCC) component. Scale bar: 300 μm (black box) and 10 μm (dark bar). **d** WB analysis of protein expressions in mouse lung SCC cells (mLSCC[LP.3]). **e** Co-staining of Flag/MKK7 and pJNK1/2 or p63 or ΔNp63 in mouse LSCC cells. Scale bar: 50 μm; Yellow arrows point out the cells expressing the exogenous Flag-MKK7; white arrows point out the cells without clear expression of the exogenous Flag-MKK7. **f** WB analysis of protein expressions in NL20 cells, an immortalized human bronchial epithelial cell line, after knockout of *LKB1* or scramble (Control) using lentiviral Cas9/gRNA with the treatment of TNFα (20 ng/mL). **g** WB analysis of protein expressions in NL20 cells after knockout of *MKK7* or scramble (Control) using lentiviral Cas9/gRNA under 5-h treatment of TNFα (20 ng/mL). **h** MTS assay analysis of cell viability of NL20. −: parental cells; gControl: gRNA targeting noncoding region; gLKB1: gRNA targeting LKB1; gMKK7: gRNA targeting MKK7

| Table 4 Ingenuity pathway analysis (IPA) of the differentially expressed genes after knockout *Jnk1/2* in MSCC[LP.3] cells (Supplementary Data 3) | |
| --- | --- |
| | **P value** |
| Top diseases | |
| Cancer | 2.25E-06–5.73E-19 |
| Top cellular function | |
| Cellular growth and proliferation | 2.25E-06–2.12E-28 |
| Cell death and survival | 2.25E-06–3.84E-26 |
| Enriched upstream regulators | |
| TNF | 1.26E-30 |
| TP63 | 8.77E-20 |
| IL1B | 2.33E-18 |

## Discussion

In this study, we generated a mouse lung SCC model in which the sole ablation of *Lkb1* in mouse lung airways is sufficient to induce LSCC and have established *Lkb1* as a key suppressor gene of LSCC. *Lkb1* deficiency results in decreased expression of MKK7, a reduction of JNK1/2 phosphorylation, lower JNK1/2 activities and elevated ΔNp63 signaling, which subsequently leads to epithelial cells transformation and progression into LSCC (Fig. 7). Loss of *Lkb1* however does not automatically cause loss of p-JNK1/2 in normal mouse airway epithelial cells. This would indicate that other events must occur to alter p-JNK1/2 levels. However, even though p-JNK1/2 is lost during tumor progression, JNK1/2 protein levels are maintained. Therefore, JNK1/2 activities can still be stimulated at the tumor stage (Figs. 4e and 5a) and activation of JNK1/2 signaling may serve as a tool against LSCC for both treatment and prevention.

Previous studies showed that simulation of LSCC in vivo had only been successfully achieved by compound mutations, including *Kras*[G12D]/*Lkb1*[loss], *Sox2*[OX]/*Lkb1*[loss], *Pten*[loss]/*Lkb1*[loss], and *Cdkn2a*[loss]/*Pten*[loss]/*Sox2*[OX8–11]. Although *Lkb1* is involved in three of these mouse models, ablating *Lkb1* alone using adenovirus-Cre (Ad-Cre) fails to cause pulmonary neoplasia[9]. As a result, *Pten* loss or *Sox2* overexpression was given more weight as the major factors in driving LSCC development. Since the efficiency and accuracy of delivering Cre recombinase intranasally and intratracheally remain uncertain[27,48], findings based on this method may be limited in scope due to technical challenges, especially for pathogenesis in the mouse large airways where human LSCC is frequently initiated[5]. On the contrary, *CCSP*[iCre] is a proven efficient Cre driver that mediates the loxP-dependent DNA recombination in all airway epithelial cells including the large airways[13,49]. Results based on the *CCSP*[iCre] system indeed revealed a previously underestimated role of *Lkb1* lost-of-function mutation that by itself *Lkb1* deficiency is sufficient to initiate LSCC, in comparison with any single mutation in either

*Trp53*, *Pten*, *Errfi1*, *Smad4* or *Kras*[G12D] (Table 1)[10,12–14]. Furthermore, the merit that *Lkb1* ablation alone induces LSCC offers a simplified platform for screening other regulators of LSCC development. These observations not only place *Lkb1* in its right place of the LSCC driving formula, but also show that the *CCSP*[iCre]*Lkb1*[f/f] model is a valuable tool for further investigating additional players that contribute to LSCC development.

In the *CCSP*[iCre]*Lkb1*[f/f] model, LSCC and lung adenocarcinoma were formed at different locations of mouse lung. LSCCs were usually located in the central part of the lung (Fig. 1a and Supplementary Fig. 1c), close to the trachea, after the deletion of *Lkb1* in large airways (Supplementary Figs. 1a–b and 5a–b). In contrast, lung adenocarcinomas were often found in the distal parts of *CCSP*[iCre]*Lkb1*[f/f] mouse lung (Supplementary Fig. 2a). This is consistent with human LSCC that belongs to central lung neoplasm[50] and is frequently initiated from large airways[5]. Taken together, our *CCSP*[iCre] mouse can be an efficient model for the in vivo study of lung diseases initiated from large airway and of the de novo role of these frequently mutated genes.

JNK1/2 inactivation is associated with not only LSCC but also cervical and head and neck SCC (Fig. 6), suggesting a conserved role of JNK1/2 in various types of SCC development. JNK1/2 activation is required to induce cell apoptosis in multiple SCC cells, including esophageal, oral, head and neck and cervical SCC cells[44–47]. These results suggest that JNK1/2 inactivation in vivo may promote or lead to SCC formation and progression besides lung SCC. A recent in vivo example is the off-target actions of BRAF inhibitors. They were used for inhibiting melanoma growth, but at the same time also promoted squamous cell carcinoma development that has been reported due to JNK pathway inhibition[51]. In addition to lung SCCs, *LKB1* and *MKK7*, as upstream regulators of P-JNK1/2, are also found to be frequently mutated in other human SCCs, such as cervical, cutaneous, head and neck and esophageal SCCs (cBioPortal databases)[52,53]. Meanwhile, we showed that the ΔNp63 pathway, a well-proven oncogenic and squamous linage pathway[36], was activated after JNK1/2 inactivation. This further suggests that JNK1/2 inactivation promotes SCC development through a common downstream target p63 since TP63 was also amplified in human cervical and head and neck SCCs (cBioPortal databases). In summary, JNK1/2 pathway may act as a conserved gatekeeper preventing the development of SCCs.

Current targeted therapies for LSCC are limited because the key drivers have not been identified. Although the *Ikka* mutant (K44A) knock-in mice does develop spontaneous LSCCs[54], K44A mutation has not been identified in human LSCC, and the copy number losses and other genomic mutations in *IKKa* are also rare in human LSCCs[7]. Here, we demonstrated the potential to target JNK1/2 for human SCC treatment in preclinical models in that activation of P-JNK1/2 using Anisomycin induced LSCC cell death and inhibited LSCC development (Fig. 5). Anisomycin, a natural product produced by some Streptomyces bacteria,

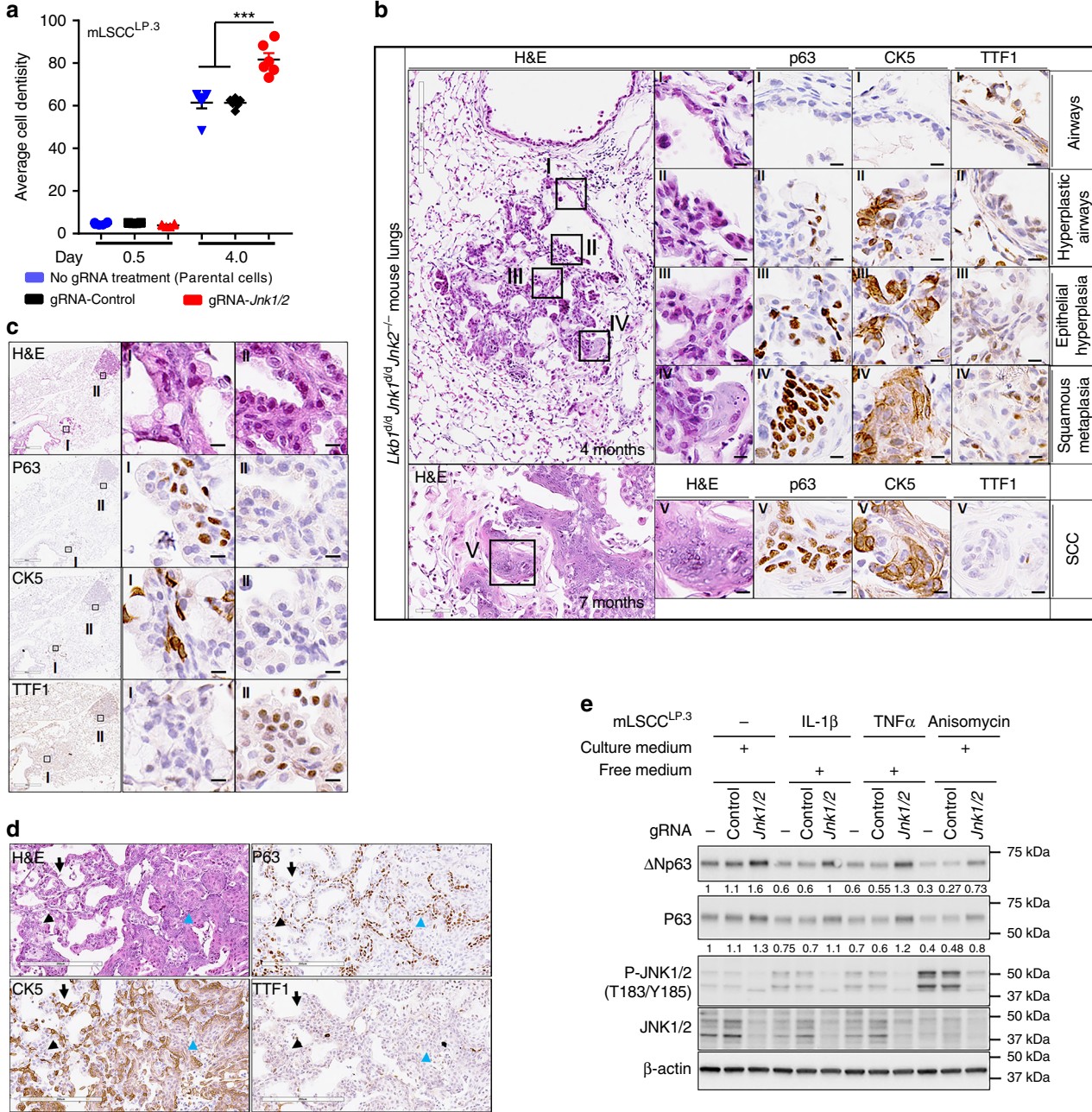

**Fig. 4** *Jnk1/2* ablation accelerated *Lkb1*-deficiency-induced LSCC progression. **a** Quantification of cell colony formation assay of mLSCC[LP.3] cells after knockout of *Jnk1/2* or scramble (Control) using Lentiviral Cas9/gRNA. Error bar is SEM; *T* test: ***$P < 0.001$. **b** Representative H&E and IHC staining of 4-month-old (4M) and 7-month-old (7M) *CCSP*[iCre]*Lkb1*[f/f]*Jnk1*[f/f]*Jnk2*[−/−] (*Lkb1*[d/d]*Jnk1*[d/d]*Jnk2*[−/−]) mouse lungs. SCC markers: p63 and CK5; AD marker: TTF1. Scale bar: 600 μm (black box, 4 months), 60 μm (black box, 7 months), and 10 μm (solid black). **c** Representative H&E and IHC staining of 7-month-old *Lkb1*[d/d]*Jnk1*[d/d]*Jnk2*[−/−] mouse lung epithelial hyperplasia (I), and adenocarcinoma (II). Scale bar: 300 μm (black box, left panel), 60 μm (black box, middle and right panels). **d** The representative H&E and IHC staining of lung adenosquamous carcinoma in 7-month-old *Lkb1*[d/d]*Jnk1*[d/d]*Jnk2*[−/−] mice. Adenocarcinoma (AD) cells are indicated by black arrows, and squamous cell carcinoma (SCC) cells in the lower half of the image are indicated by the green arrowhead. A large portion of the AD component has positive staining for the typical SCC markers CK5 or p63 while having negative or weak staining for TTF1. The SCC component has positive staining for CK5 and p63 while having negative staining of TTF1. Scale bar: 200 μm. **e** WB analysis of protein expressions in mLSCC[LP.3] cells under 5-h treatment of IL-1β (2 ng/mL), TNFα (20 ng/mL) and Anisomycin (10 μM). Free medium and culture medium are described in the Methods section

stimulates JNK1/2 phosphorylation[55]. Clinically, Anisomycin has been successfully used for the treatment of amoebic dysentery and trichomoniasis[55]. It also sensitizes tumor cells to apoptosis induced by the TNF-related apoptosis-inducing ligand (TRAIL)[56]. The observation that JNK1/2 activities positively correlate with the survival rates of multiple types of SCC patients suggests

application of JNK1/2 activators, such as Anisomycin or its derivatives could be an option for SCC patients with low JNK1/2 activities.

Drugs targeting JNK1/2 that are currently under clinical trial or FDA-approved for the treatment of other cancers could be investigated for their anticancer effects on LSCCs. For

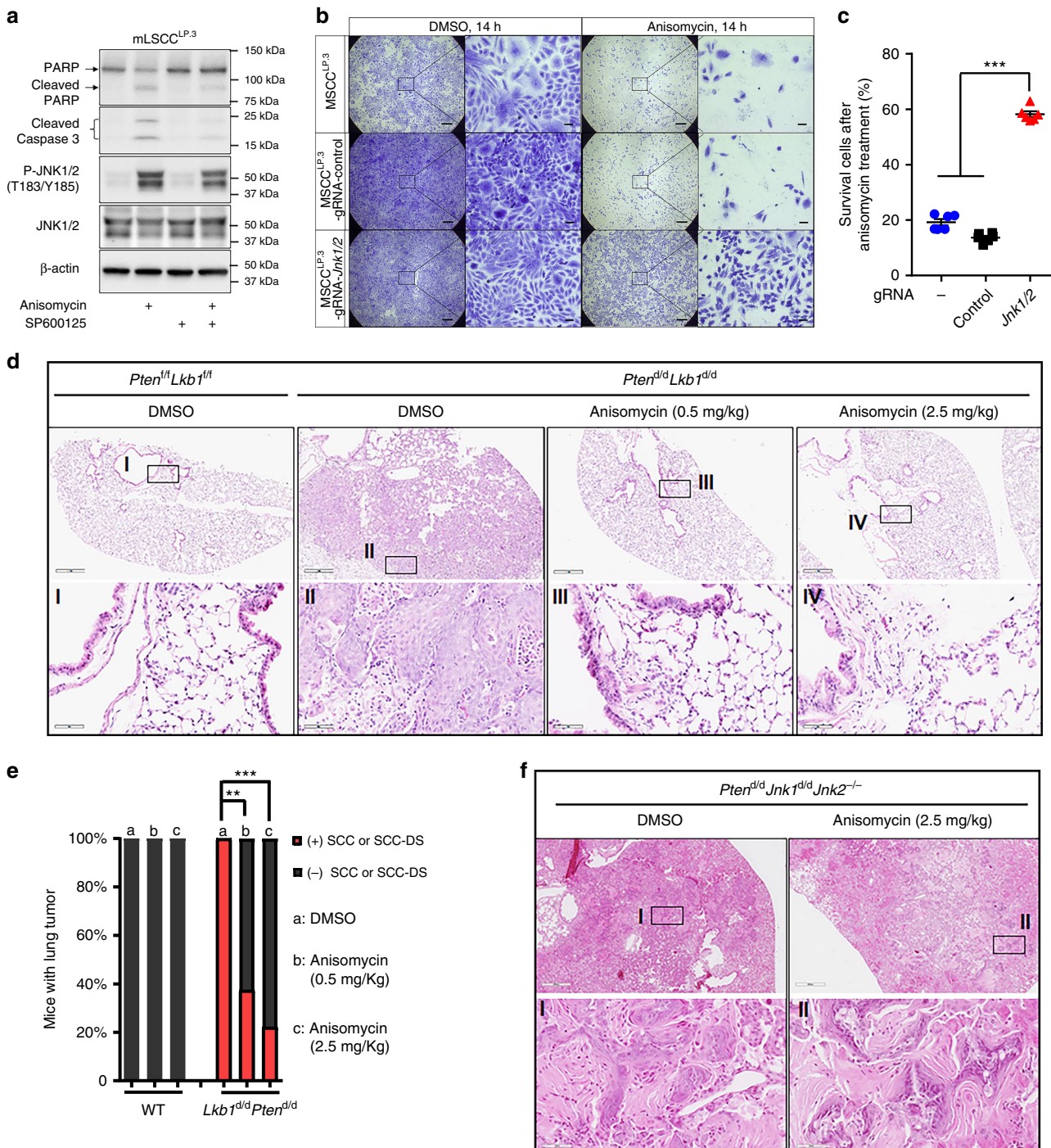

**Fig. 5** Pharmaceutical activation of JNK1/2 attenuates LSCC development. **a** WB analysis of protein levels in mLSCC$^{LP.3}$ cells with the treatment of Anisomycin (10 μM, 5 h) and SP600125 (10 μM, 4.5 h). **b** Representative crystal staining of survival mLSCC$^{LP.3}$ parental cells and stable cells with lentiviral Cas9/gRNA targeting Control or *Jnk1/2* after treatment with Anisomycin or DMSO. Scale bar: 500 μm (large area) and 50 μm (small area). **c** Quantification of cell survival assay of survival mLSCC$^{LP.3}$ parental cells and stable cells with lentiviral Cas9/gRNA targeting Control or *Jnk1/2* was performed. Error bar is SEM; ANOVA-TKMC Test: ***$P < 0.001$. **d** Representative H&E staining of lungs or lung tumors of the mice treated with DMSO or Anisomycin. **e** Quantification of mice with lung SCC or SCC-DS at the endpoint of treatment (Fig. 5d). The b group of *Lkb1*$^{d/d}$*Pten*$^{d/d}$ had eight mice and the remainder of each group included nine mice. $\chi^2$ test (two-way): **$P < 0.01$; ***$P < 0.001$. **f** Representative H&E staining of lungs or lung tumors of the mice treated with DMSO ($n = 9$) or Anisomycin ($n = 10$)

example, Genistein, which is an enzyme inhibitor and in clinical trial, has been showed to have antitumor effects in human hepatocellular carcinoma cells and to activate JNK1/2 [57]. In addition, 2-methoxyestradiol, which belongs to mitosis modulators and is also in clinical trial, induced apoptosis in prostate

cancer cells through the activation of P-JNK1/2[58]. Therefore, repurposing existing drugs to activate JNK1/2 for LSCC treatment is possible.

Although epithelial hyperplasia and squamous metaplasia are regarded as the initial stages of human LSCC[5], not much was

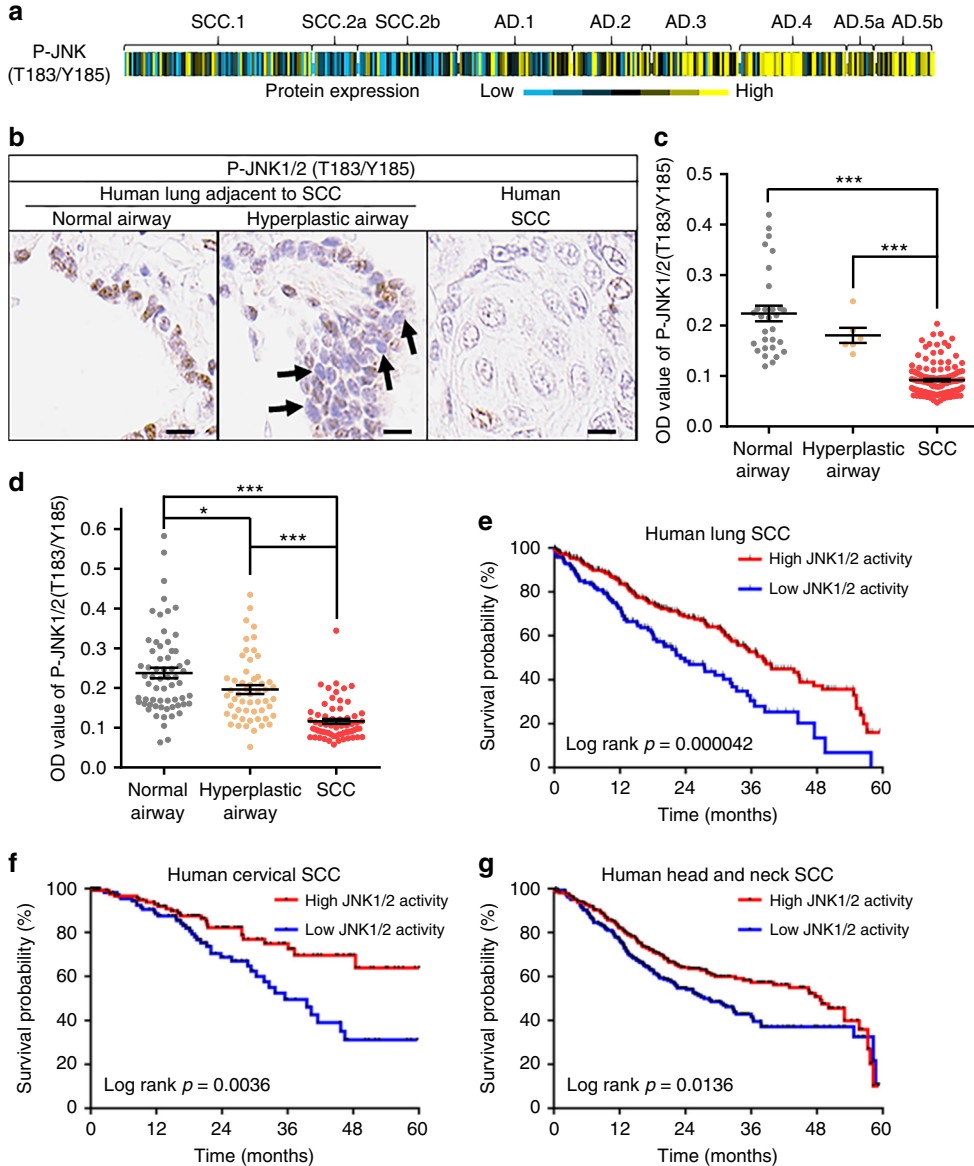

**Fig. 6** JNK1/2 is inactivated in human LSCCs and predicts SCC patients' survival. **a** The heatmap of the reverse phase protein array (RPPA) value of P-JNK1/2 in TCGA human lung cancer cohorts (SCC.1, $n = 159$; SCC.2a, $n = 38$; SCC.2b, $n = 88$; AD.1, $n = 94$; AD.2, $n = 68$; AD.3, $n = 75$; AD.4, $n = 90$; AD.5a, $n = 25$; AD.5b, $n = 49$). **b**, **c** Representative IHC staining (**b**) and the semi-quantification (**c**) of P-JNK1/2 in a human LSCC tissue array. The quantifications were calculated with DAB staining density (brown color) using ImageJ. Optical Density (OD) = log (max intensity/Mean intensity). Two random equal regions of each patient were quantified for SCC (70 patients), and 30 normal airway regions (eight patients) and eight hyperplastic airways (four patients) were quantified. ANOVA (TKMC test) was performed and the error bar is SEM; ***$P < 0.001$. Arrows (left panel): Weak nuclear staining; Scale bar: 10 μm. **d** Semi-quantification of a tissue array was calculated with DAB staining density (brown color) as described in (**c**); three random equal regions of each patient were quantified, including SCC (21 patients), the matched normal airway (21 patients) and the hyperplastic airways (18 patients). ANOVA (TKMC Test) was performed and the error bar is SEM; *$P < 0.05$; ***$P < 0.001$. **e** Prediction of human LSCC patient survival rate using *Jnk1/2* knockout mouse signature. The gene signature was generated using the DEGs identified after knockout of *Jnk1/2* in MSCC^LP.3 cells (Supplementary Data 3). High JNK1/2 activities were defined as patient gene signature scores lower than 0 and vice versa. Each small bar of curve line represents one patient and *T* test was performed. If the score was below 0, JNK1/2 activities were considered high, and vice versa. **f**, **g** Higher JNK1/2 activities are associated with longer survival rate of cervical and head and neck SCC patients. The method was as described in (**e**) and Methods

described in previous in vivo models[4,11]. Hence, the mechanism of LSCC formation and progression regulated by genetic alternations are largely unknown. Our work documented tumor progression at the stages of epithelial hyperplasia and squamous metaplasia with different clinical markers as well as provided underlying mechanisms that contribute to pathogenesis. First, we observed epithelial hyperplasia with squamous metaplasia had positive expression of p63 and CK5 in *Lkb1*^d/d (Supplementary

Fig. 1e) and in *Lkb1*^d/d*Jnk1*^d/d*Jnk2*^−/− (Fig. 4b) mice, but both epithelial hyperplasia and adenomas had negative expression of p63 and CK5 in *Kras*^G12D mice (Supplementary Fig. 2f). Consistent with the observations in human lung cancers, these mouse results show that epithelial hyperplasia can develop into SCC, AD or ASC, and the developmental direction of epithelial hyperplasia can be reflected by expression of these markers, such as p63 and CK5 for SCC and TTF1 for AD. In the ASCs, the

portions of tumors consisting of AD cells also exhibited the same marker expression of SCC cells (p63 and CK5) in $Lkb1^{d/d}$ (Supplementary Fig. 1g), in $Lkb1^{d/d}Pten^{d/d}$ (Supplementary Fig. 3e) and in $Lkb1^{d/d}Jnk1^{d/d}Jnk2^{-/-}$ (Fig. 4d) mice. Meanwhile, ASC's molecular profiles are almost identical to that of SCC (Supplementary Fig. 3g). Therefore, the clinical marker staining, such as p63 and CK5[59], can predict the developing tract before the formation of these typical pathological phenotypes. Second, $Lkb1^{d/d}$, $Lkb1^{d/d}Jnk1^{d/d}Jnk2^{-/-}$ and $Lkb1^{d/d}Pten^{d/d}$ mouse models in our study can recapitulate the different LSCC developmental stages (DSs) (Fig. 4b–d; Supplementary Figs. 1d–g and 3d–e; Supplementary Data 1–2). This offers the possibility of systematically analyzing the dynamic changes of molecules during LSCC development and to identify these genetic "gatekeepers" of LSCC progression in vivo. We observed that the downregulation of P-JNK1/2 was associated with the progression of LSCC in $Lkb1^{d/d}$ and $Lkb1^{d/d}Pten^{d/d}$ mice. As we described in Table 5, $Jnk1/2$ knockout under $Lkb1$-loss background not only shortened the average developing time of LSCC-DSs from 13 months to 4 months, but also increased the incidence of LSCC-DSs from 25 to 100%, besides the promotion of LSCCs. Unlike $Lkb1^{d/d}$ mouse lungs in which only some of epithelial hyperplasia were positive for staining of p63 and CK5 (Supplementary Fig. 1c–d and Supplementary Data 2), they can be detected in all epithelial hyperplasia of $Lkb1^{d/d}Jnk1^{d/d}Jnk2^{-/-}$ mouse lungs (Fig. 4b, c and Supplementary Data 2). Furthermore, some of lung adenocarcinoma in $Lkb1^{d/d}Jnk1^{d/d}Jnk2^{-/-}$ mice showed p63 and CK5 positive staining (Supplementary Fig. 6f). These results conclude that JNK1/2 inactivation drives LSCC formation and development in response to $Lkb1$ deficiency. Therefore, evaluation of LSC-DSs can facilitate studying what and how genetic elements regulate LSCC development.

Investigation of how JNK1/2 signaling inhibits ΔNp63 expression helps further dissect the mechanism of JNK1/2 repressing LSCC development. One potential link between JNK1/2 and ΔNp63 is c-Jun. ΔNp63 has been reported to be

negatively regulated by c-Jun in response to Amyloid-β-Induced cell stress[60] while JNK1/2 is a typical activator of c-Jun[34]. Interestingly, our microarray analysis of SCCs in $Lkb1^{d/d}$ and $Lkb1^{d/d}Pten^{d/d}$ mice showed that c-Jun expression is significantly decreased (Supplementary Data 3). We also observed the significant reduction of c-Jun in a $Jnk1/2$ knockout array (gRNA-Jnk1/2 vs. gRNA-Control) (Supplementary Data 3). These results suggest inactivation of the JNK1/2 pathway induces ΔNp63 expression partially due to the decrease of c-Jun. Continued identification of the pathways regulated by JNK1/2 will serve to better target and customize therapy for SCC.

The LSCC model resulting from $Lkb1$ loss in the pulmonary epithelium represents a model to study the cellular origins of SCCs. Anatomic location precursors suggest that SCCs in our mouse model arise de novo (Figs. 1b and 4b; Supplementary Fig. 1c-e and Supplementary Data 2). Our $CCSP^{iCre}$ has been shown to have Cre activity in the basal cells of the proximal airway[13], which are p63-positive and are generally thought to be one of the main cellular origins[11]. This suggests that some of SCC cells were developed from these p63-positive basal cells. The observation of SCC only in $Lkb1^{d/d}$ (8.9%), $Lkb1^{d/d}Pten^{d/d}$ (11.1%) and $Lkb1^{d/d}Jnk1^{d/d}Jnk2^{-/-}$ (16.7%) mice also supports this idea (Supplementary Data 1-2). Here, we defined the mice of SCC only as the ones that have only SCC lesions and non-adenocarcinoma precursors. Meanwhile, SCCs may also transdifferentiate from adenocarcinoma or its precursors. This idea was indicated by the SCC marker staining (CK5 and P63), which was positive in SCC-DSs of these models (Supplementary Figs. 1f–g, 3e and 6e–f; Fig. 4c, d and Supplementary Data 2), especially for these adenocarcinoma precursors. Considering the reports of the trans-differentiation from $Lkb1$-deficient adenocarcinoma (AD) into SCC in Ad-Cre-$Kras^{G12D}Lkb1^{f/f}$ mice[37,61,62], this possibility also exists. The use of single-cell RNA-Seq tumors for this model at varying stages of development will help identify the changes in cell types and the transcriptome of these cells during SCC progression. This will aid in understanding the cellular origins of SCC.

Interestingly, the COSMIC database (https://cancer.sanger.ac.uk/cosmic), which includes 1588 SCC and 36 ASC samples among 8018 patients, shows that the frequency of $LKB1$ mutations in pure human SCC and ASC is 1.89% and 11.11%, respectively. In other words, the mutation rate in ASC is much higher (5.89 fold) than SCC. Therefore, $LKB1$ may require secondary mutations, such as $PTEN$ or alterations in JNK1/2 signaling to direct the progression of the de novo tumors to SCC vs. ASC or to promote the progression of ASC to SCC. Meanwhile, we observed one common mutation (CDS: c.1062C > G; AA: p. F354L) between SCC and ASC, which may indicate the potential for transdifferentiating.

In summary, we exploited the experimental merits of the mouse to establish $Lkb1$ as a key driver gene from those frequently mutated genes in human LSCCs, and to understand the formation and progression of LSCC by unveiling the de novo role of JNK1/2 inactivation. Extending the finding of JNK1/2

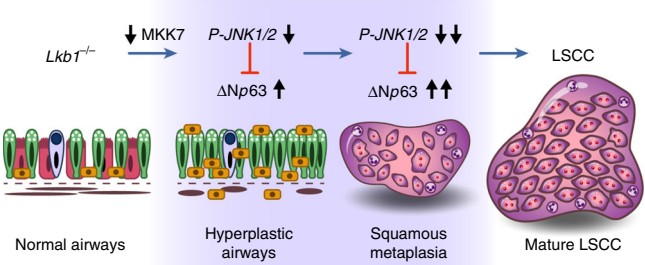

**Fig. 7** Working model. $Lkb1$ loss in airways decreases MKK7 expression levels, reduces JNK1/2 phosphorylation levels and activities, and de-represses oncogenic and squamous linage pathway(s) (e.g. ΔNp63 pathway), which prompts hyperplasia and squamous metaplasia in normal airway cells, breaks the barrier against LSCC formation, and promotes LSCC development

**Table 5 Summary of $Lkb1^{d/d}Jnk1^{d/d}Jnk2^{-/-}$ mouse lung tumor phenotypes in comparison to $Lkb1^{d/d}$ mouse lung tumor phenotypes**

| Genotype | No. | Age (Month) | The percent (%) of phenotypes in mouse lungs | | | | | |
|---|---|---|---|---|---|---|---|---|
| | | | SCC-DS | SCC | SCC-DS + SCC | AD-DS | AD | AD-DS + AD |
| $Lkb1^{d/d}$ | 56 | 11–14 | 25.0 | 16.1 | 32.8 | 16.1 | 37.5 | 53.6 |
| $Lkb1^{d/d}Jnk1^{d/d}Jnk2^{-/-}$ | 12 | 7–8 | 100 | 33.3 | 100 | 8.3 | 33.3 | 33.3 |

SCC squamous cell carcinoma, DS development stage, AD adenocarcinoma, No. number

from mice to human shows that JNK1/2 is inactivated in a large population of LSCCs and its activity is positively associated with longer survival rate of multiple human SCC patients, including LSCC. Our model-informed progression analysis, together with genetic, functional and clinical relevance studies, establishes JNK1/2 as a key regulator of LSCC progression in mice and humans, providing a promising target for the therapy of SCC patients with low JNK1/2 activities, including LSCCs.

## Methods

**Mice**. CCSP[iCre] and Mig6[f/f] mouse strains were made by Dr. DeMayo's lab and Jnk1[f/f]Jnk2[−/−] mouse strain was made by Dr. Roger Davis's lab and the generation methods of these mice were in previous publications[13,63,64]. Lkb1[f/f] (FVB;129S6-Stk11[tm4Rdp]/Nci), Kras[G12D] (B6.129-Kras[tm4Tyj]/Nci) and p53[f/f] (FVB.129P2-Trp53[tm1Brn]/Nci) mouse strains were from the NCI Frederick Mouse Repository. Pten[f/f] (C;129S4-Pten[tm1Hwu]/J) strain were from Jackson Lab. Smad4[f/f] (Smad4[RobCA]) stain is from Dr. Elizabeth J. Robertson's lab. The deletion or over-expression of those genes in mouse lung was achieved by breeding conditional genetic mice with CCSP[iCre] mice. All animal protocols were approved by the National Institute of Environmental Health Sciences and Baylor College of Medicine. All experiments were conducted in accordance with relevant guidelines and regulations of both institutions. All strains were B6;129 and all mice were genotyped by Transnetyx. Similar numbers of female and male mice were purposely included in all the mouse experiments, and no histopathologic differences were observed between female mice and male mice during lung tumor development.

**Anisomycin treatment**. WT (Lkb1[f/f]Pten[f/f]) and Lkb1[d/d]Pten[d/d] (CCSP[iCre]Lkb1[f/f]Pten[f/f]) mice at 6 weeks old received Anisomycin (SIGMA, A9789) or vehicle through intraperitoneal (i.p.) injection for 8 weeks[65,66]. Anisomycin was dissolved in DMSO and diluted in PBS, and the final dissolved reagent was constituted by 7.5% DMSO and 92.5% PBS, which served as the vehicle. Two concentrations (0.5 mg/kg and 2.5 mg/kg) of Anisomycin were used for treatment three times per week[67]. Similarly, Lkb1[d/d]Jnk1[d/d] Jnk2[−/−] (CCSP[iCre]Lkb1[f/f]Jnk1[f/f]Jnk2[−/−]) mice at 6–7 months old received Anisomycin (2.5 mg/kg) or vehicle through intraperitoneal (i.p.) injection for 8 weeks. Fifty microliters was used per injection. Two hours after the last treatment, mice were euthanized for tissue collection and examination of lung.

**Cell derivation and culture**. mLSCC[LP.3] cells were isolated from Lkb1[d/d]Pten[d/d] mouse lung SCC tumors using a cancer cell isolation kit as per the protocol. NL-20, Beas-2B and H157 cells were purchased from ATCC. The culture medium for MSCC[LP.3] cells included RPMI, 10% FBS, penicillin (100 IU/mL) and streptomycin (100 μg/mL). NL20 was cultured using complete growth medium: Ham's F12 medium, 2.7 g/L glucose, 0.005 mg/mL insulin, 10 ng/mL epidermal growth factor, 0.001 mg/mL transferrin, 500 ng/mL hydrocortisone and 4% FBS. Beas-2B was cultured using the BEGM kit (Lonza, No. CC-3170). H157 cells was cultured using DMEM and 10% FBS, penicillin (100 IU/mL) and streptomycin (100 μg/mL). Given that P-JNK1/2 in lung SCC tumors of Lkb1[d/d]Pten[d/d] mice was low (Fig. 2c, d), TNFα or IL-1β was used to treat mLSCC cells. Before the treatment of TNFα or IL-1β, the cells were starved with free medium overnight. The free medium means RPMI for mLSCC[LP.3] cells or Ham's F12 medium for NL20 cells or BEBM medium for Beas-2B cells. As described in the American Type Culture Collection (ATCC), an NL20 cell is an immortalized and nontumorigenic human bronchial epithelial cell line that was established by expression of replication-defective SV40 large T antigen, and Beas-2B cells were isolated from normal human bronchial epithelium obtained from the autopsy of noncancerous individuals and infected with an adenovirus 12-SV40 virus hybrid (Ad12SV40) and cloned. H157 cells are human LSCC cells with mutations of LKB1 and PTEN.

**Plasmid transfection**. Flag-MKK7 and Flag-pcDNA3.1 were purchased from Addgene. Both plasmids were transfected into mLSCC[LP.3] and H157 cells following the manufacturer protocol of Lipofectamine 2000. Western blot and immuno-fluorescence (IF) were performed to examine the transfected cells after transfection for 48 h using the similar methods[68]. Details of IF staining process: 1. Place cells on 12 mm coverslips in 24-well plates (Fisher-12-545-82-12cir-1D); 2. Wash cells twice using 1× PBS (pH 7.4); 3. Fix cells in 0.5 mL 4% paraformaldehyde (10 mL 16% in 30 mL PBS) (polysciences, Inc. # 18814) for 10 min at RT; 4. Wash cells twice using 1× PBS (pH 7.4); 5. Incubate cells in 0.5 mL 0.5% Triton-X-100 in PBS for 5 min at RT; 6. Wash cells twice using 1× PBST (PBS with 0.2% Tween 20, pH 7.4); 7. Prepare blocking buffer using 1× PBST with 5% normal donkey serum (Jackson Immune Research 017-000-121) plus 0.2% Fish Stain Gelatin (leave at 37 °C to dissolve), and incubate cells in blocking buffer 60 min at room temperature (RT); 8. Incubate cells using primary antibodies for overnight at 4 °C, which is diluted in blocking buffer (1:100–1:1000); 9. Wash cells three times using 1× PBST 5 min each at RT; 10. Incubate cells using secondary Ab for 60 min at RT, which was diluted in blocking buffer (1:200); 11. Wash cells three times using 1× PBST 5

min each at RT; 12. Place cells on slide using Vectashield Mounting Media (H1500, Hardset); 13. IF pictures were taken using confocal microscopy.

**Cell proliferation, colony and survival assay**. For cell proliferation assay, ~2000 NL20 and Beas-2B cells were seeded in one well in a 96-well plate at day 0, respectively. Each group has four replicates. Each assay was performed by adding 20 μL MTS reagent (Promega, #G3582) into each well containing 200 μL cell culture medium, incubating for 1 h and then recording absorbance at 490 nm with a 96-well plate reader. mLSCC[LP.3] cells were cultured in six-well plates and each experiment group included six wells for colony and survival assay. For cell colony assay, ~10,000 cells were seeded at day 0. For the survival assay, ~1 × 10⁵ cells were seeded at day 0. At the endpoint of experiments, cells were washed one time using 1× PBS and then fixed by 3.7% paraformaldehyde (PFA) for 15 min at room temperature (RT). PFA was removed and the cells were stained for 30 min with 0.05% Crystal Violet[69]. Cells were gently washed five times using tap water. Plates were drained by inverting them overnight, and each well was photographed. The stained cells were counted based on the photo using ImageJ method[70].

**Histopathology and immunohistochemistry**. The experiments were performed following the similar methods[8,17]. Mouse lungs were fixed in 4% paraformaldehyde and paraffin-embedded according to standard protocols. H&E and IHC analyses were performed on 5 μm lung sections. Sections underwent dewaxing using CitriSolv clearing agent followed by antigen retrieval and peroxide quenching with 3% $H_2O_2$ in methanol for 15 min and blocked with 5% normal goat serum plus Avidin (four drops of Avidin from Avidin/Biotin Blocking Kit/mL) or MOM blocking reagents at RT for 1 h. Subsequently, the slides were incubated with primary antibodies diluted in 5% normal goat serum plus Biotin (four drops of Biotin from Avidin/Biotin Blocking Kit/mL) at 4 °C overnight. Slides were washed using 1× PBS five times (5 min per time), and then incubated with 1% BSA to block at RT for 30 min and then followed with secondary antibodies diluted in 1% BSA at RT for 30 min. The slides were then developed using the Vectastain ABC kit and DAB reagents. The subgroups of different proteins from the same staining slide (e.g., four H&E groups (Supplementary Fig. 1d) are indicated in the large area (Supplementary Fig. 1c). All H&E and IHC results were scanned at the NIEHS Image Core, including the tissue array slides. Scanning method: 1. Slides were cleaned with an isopropanol solution prior to scanning. 2. Slides were then scanned using the Aperio AT2 Scanner, (Aperio: Leica Biosystems Inc., Buffalo Grove, IL). This machine uses line scanning technology to capture high-resolution, seamless digital images of glass slides. 3. After scanning, slides were viewed and captured using Aperio ImageScope v. 12.3.0.5056 (Aperio: Leica Biosystems Inc., Buffalo Grove, IL), a digital slide viewing program. Measurements were taken from these images using a measurement scale that was internally calibrated for each image as it was scanned. p63 and ΔNp63 and TTF1 are nuclear staining and multifocal. Squamous cell carcinomas showed positive staining of both P63 and ΔNp63. Within the SCCs, the compact arrangements of basal-appearing cells with scant cytoplasm tended to show nuclear staining of p63.

**Human tissue array**. The array samples shown in Fig. 6b, c and Supplementary Fig. 8a were purchased from US Biolab. The detailed patient information can be downloaded from US Biolab website. And the other batches of array (Fig. 6d and Supplementary Fig. 8b) were from the Pathology and Histology Core at Baylor College of Medicine. Histological examinations were confirmed by Dr. Patricia D. Castro and Dr. Michael M. Ittmann. The detailed patient information can be found in Supplementary Table 8. The full staining slides were scanned at the NIEHS Image Core as described in the paragraph of Histopathology and IHC, and random regions of each patient were quantified using the optical density (OD) values, which were calculated based on the DAB staining density (brown color) using ImageJ[17]. OD = log (max intensity/mean intensity). ANOVA (TKMC Test) was performed.

**Microarray and ingenuity pathway analysis**. *RNA isolation*: Following the same RNA isolation protocol[8], total RNAs were isolated from mouse lungs or lung tumors using TRIzol reagent and cleaned by RNeasy kit. For mouse cells, total RNAs were isolated by RNeasy kit. All mRNAs were reversely transcribed into cDNA with the M-MLV kit. RNA quality was assessed by using the Agilent Model 2100 Bioanalyzer (Agilent Technologies, Palo Alto, CA).

*Experiment processes*: (1) GSE111313 and GSE111314 arrays were performed at NIEHS. Gene expression analysis was conducted using Agilent Whole Mouse Genome 4 × 44 multiplex format oligo arrays (Agilent Technologies, 014868) following the Agilent one-color microarray-based gene expression analysis protocol. Starting with 400 ng of total RNA, Cy3-labeled cRNA was produced following the manufacturer's protocol. For each sample, 1.65 μg of Cy3-labeled cRNAs was fragmented and hybridized for 17 h in a rotating hybridization oven. Slides were washed and then scanned with an Agilent Scanner. Data were obtained using the Agilent Feature Extraction software (v12), using the one-color defaults for all parameters. The Agilent Feature Extraction Software performed error modeling, adjusting for additive and multiplicative noise. The resulting data were processed using OmicSoft Array Studio (Version 10) software. (2) GSE111335 was performed at BCM using SurePrint G3 Mouse GE 8 × 60 K Microarray Kit: The

Genomic and RNA Profiling Core first conducted Sample Quality checks using the NanoDrop ND-1000 and Agilent Bioanalyzer. Agilent One-Color Expression-Low Input Quick Amp v6.6 (Sept 2012) protocol was followed.

*Labeling protocol:* Fifty nanograms of total RNA, combined with RNA spike mix, were reverse transcribed using a T7 Primer Mix to produce cDNA. The cDNA product was transcribed using T7 RNA Polymerase, producing cyanine-3-labeled cRNA. The labeled cRNA was purified using a Qiagen RNeasy Mini Kit. Purified products were quantified using the NanoDrop spectrophotometer for yield and dye incorporation and tested for integrity on the Agilent Bioanalyzer. Six hundred nanograms of the labeled cRNA was fragmented.

*Hybridization protocol:* Approximately 480 ng of fragmented cRNA samples was loaded onto each of the Mouse G3 8 × 60 K Expression arrays. The arrays were hybridized in an Agilent Hybridization Chamber for 17 h at 65 °C with rotation at 10 rpm.

*Wash protocol:* The arrays were washed using the Agilent Expression Wash Buffers One and Two, followed by acetonitrile, as per the Agilent protocol.

*Scan protocol:* Once dry, the slides were scanned with the Agilent Scanner (G2565BA) using Scanner Version C and Scan Control software version A.8.3.1. Data extraction and quality assessment of the microarray data was completed using Agilent Feature Extraction Software Version 11.0.1.1.

*Expression array analysis:* The files with raw expression value were used to identify differentially expressed genes (DEGs) by the Genomics Suite Gene Expression workflow of Partek software package version 6.6 (Partek Inc., St. Louis, MO, USA). The quantile normalization and log2 transformation were applied to generate signal values of all samples. The expression profiles were compared among different groups using a one-way ANOVA model, and all the DEGs are listed in Supplementary Data 3.

*Functional analysis:* Functional analysis of DEGs was performed using the Ingenuity Pathway Analysis (IPA, www.ingenuity.com) based on the content of 2015-03-22. For a given biological category in IPA, Fisher's exact test was used to calculate the $P$ value. The categories with $P$ values less than 0.05 were defined as significantly enriched. Microarray data were deposited in the Gene Expression Omnibus (GEO, accession GSE111313, GSE111314 and GSE111335).

**Real-time quantitative PCR and western blotting (WB).** These experiments were done using the same conditions[8,17]. Briefly, lung tissues were stored at −80 °C and total RNA was extracted using the TRIzol method followed by purification using an RNeasy Mini Kit. RNA quality was assessed by using the Agilent Model 2100 Bioanalyzer (Agilent Technologies, Palo Alto, CA). Total RNAs were reverse transcribed into cDNA using the M-MLV kit following the protocol. The list of primers for SYBR® Green is shown in Supplementary Table 9. The Taqman probes for 18s (4310893E) and *Lkb1* (Mm00488470_m1) were purchased from Applied Biosystems. Full blots of key figures can be found in Supplementary Fig. 9.

**Kinome array.** The kinome array was purchased from R&D Systems (Cat. no.: ARY003B), which can simultaneously detect the relative site-specific phosphorylation of 43 kinases and 2 related total proteins. Each experiment group (Supplementary Fig. 4b-c and Supplementary Table 5) included six mouse lungs (~15 mg per mouse). Lungs were combined and homogenized to extract proteins. The remainder of the steps followed the standard protocol of the array kit. Non-saturated dots in the membrane were quantified using ImageJ[13] and normalized in the other groups to WT group (*Lkb1*^f/f *Pten*^f/f mouse lungs).

**Cas9/gRNA experiments.** The gRNA sequences are included in Supplementary Table 9. These gRNAs were cloned into the LentiCRISPRv2 vector[63]. CRISPR-Lenti nontargeting control plasmid was purchased from MilliporeSigma (Billerica, MA) (CRISPR12-1EA). MAX Efficiency DH5α (ThermoFisher Scientific, Cat# 18258012) were used to amplify plasmids. Lentiviruses were produced in the Viral Vector Core Laboratory of NIH/NIEHS or Baylor College of Medicine and were used to infect cells for the knockout of the specific gene. The pooled cells infected by gRNA were collected for western blot to confirm the knockout efficiency. Noninfected and gRNA-control-infected cells were used as controls.

**Gene signature analysis.** The gene expression array dataset GSE11969 (Takeuchi et al., 2006) was analyzed with the mouse lung cancer gene signature approach[13]. The publicly available datasets of human SCC ($n = 553$), AD ($n = 576$), head and neck SCC ($n = 566$), and cervical SCC ($n = 308$) cohorts were downloaded from the UCSC Xena webpage (https://xenabrowser.net/hub/). The mean-normalized gene and protein expression values were used for further analyses. The patient phenotype included survival time (OS) and longer relapse-free survival (RFS). The gene expression profile of each human sample was scored based on Jnk1/2 signature, which was generated from the mouse *Jnk1/2* knockout model using our previous published method[43]. For the head and neck SCC, and cervical SCC datasets, only the genes passing expression cutoff filter of average FPKM (Fragments Per Kilobase Million) higher than 5 are included in the analysis. The gene signature score (also known as a "t-score") was defined for each sample as the two-sided $t$ statistic comparison of the high Jnk1/2-signature genes' expression profile with the low Jnk1/2-signature genes' profile. Samples with a gene signature score (t-score) higher than 0, which share a similar gene signature profile of Jnk1/2 knockout in mLSCC^L.P.3 cells, were classified as having low JNK1/2 activities and vice versa.

**Gene set enrichment analysis (GSEA).** The oncogenic signatures (c6.all.v6.1. symbols.gmt) of the GSEA 3.0 version were used. All detected genes were pre-ranked based on their fold changes (SCC vs. WT) from high to low (Supplementary Data 3). The settings of GSEA were listed: Number of permutations (1000), Enrichment statistic (Classic), Normalization mode (Meandiv), Make detailed gene set report (True) and Seed for permutation (Timestamp).

**Quantification and statistical analysis.** Data are expressed as mean + SD or mean + SEM. The sample size ($n$) represents biological replicates. Student's $t$ test was used for comparison of two group averages. When there were more than two groups, one-way ANOVA Tukey−Kramer multiple comparisons (TKMC) test was performed. GraphPad Prism version 7.02 was used to generate the survival curves and calculate the $P$ value of log-rank (Mantel−Cox) test. Spearman correlation between gene signature and protein expression and the $P$ value was calculated using Partek Genomics Suite 6.6 software. Group size was determined based on the results of preliminary experiments and no statistical method was used to pre-determine sample size in animal studies. All histopathological results were blinded to pathologists, and the evaluation reports, including the classification of different groups, are provided by pathologists. All the bioinformatics and statistical analyses were analyzed or confirmed by bioinformaticians. The genotyping of mice was conducted by Transnetyx. All the array experiments were unbiasedly conducted and analyzed by core facilities. All qRT-PCR experiments were performed in a blinded manner. The rest of group design and outcome analysis were not performed in a blinded manner. Statistical significance was considered for all the datasets when the $P$ value was less than 0.05.

**Reporting summary.** Further information on research design is available in the Nature Research Reporting Summary linked to this article.

## Data availability
Microarray data were deposited in the Gene Expression Omnibus (GEO, accession GSE111313, GSE111314 and GSE111335). Different detailed data are provided in Supplementary Data 1−5. All detailed information of key reagents, including the dilution of antibodies, was listed in Supplementary Data 5. All other data supporting the findings of this study are available from the corresponding author upon reasonable request.

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

## Acknowledgements

We appreciate Janet DeMayo for the help of proofreading the manuscript. We would like to thank Douglas Joubert, NIH Library Editing Service, for reviewing the manuscript. Many thanks to Linwood Koonce for his great mouse work in supporting this project. We appreciate Drs. Paul Wade, Steve Kleeberger, Kenneth S. Korach, Carmen J. Williams, Humphrey Yao, Guang Hu, and Robert C. Sills for their valuable suggestions. We appreciate the support from the NIEHS animal facility (Galo A. Defaz, Angela Dickerson, and Molly Comins), Knockout Mouse Core (Dr. Manas K. Ray, Dr. Artiom Gruzdev, and Gregory J. Scott), Digital Imaging Core (Eli Ney), Molecular Genomics Core (Drs. Kevin Gerrish and Liwen Liu) and Viral Vector Core (Drs. Negin Martin and Shih-Heng Chen). We also thank Image Associates, Inc. at NIEHS for the help in generating the cartoon of working model. We thank BCM Genetically Engineered Mouse Core, Biostatistics and Informatics Group, Tissue Culture, and Genomic Profiling Core. Some

human samples were provided by the Pathology and Histology Core at Baylor College of Medicine with funding from the NIH (NCI P30-CA125123). This work was mainly supported by research grants to F.J.D. (NIEHS, Z1AES103311-01).

## Author contributions

J.L. and F.J.D. designed experiments and research aims. J.L. conducted experiments with the help of M.R. and S.-N.C., and analyzed data with help from T.W. and J.-L.L.. J.L., S.-P.W. and F.J.D. wrote the manuscript with help from T.W., C.J.W. and R.J.D.. T.W., C.J.C. and J.-L.L. performed and confirmed all the bioinformatics and statistical analyses. K.S.J., C.J.W., P.D.C. and M.M.I. provided histopathological evaluations and confirmed the interpretation of tumor pathologic phenotypes. R.J.D. provided the $Jnk1^{f/f}Jnk2^{-/-}$ strain and contributed critical information and helpful discussions. F.J.D. supervised the whole project.

## Additional information

**Competing interests:** The authors declare no competing interests.

