## [Peer Review File · Nature Communications]

Reviewers' comments:

Reviewer #1, Expertise: JNK pathway

This paper examined the molecular basis of lung squamous cell carcinoma (LSCC) formation and progression. The authors found that LKB1 (serine/threonine kinase 11), a frequently mutated gene in human LSCC, acts as a tumour suppressor in the lung. More specifically, "LKB1 deficiency by itself is sufficient to induce LSCC". They further demonstrate that *Lkb1* loss of expression caused a defect in MKK7-JNK signaling, leading to increased Δ Np63 expression. The clinical implication of these findings is demonstrated by evidence that "LSCC patients with higher JNK1/2 activities have better survival rates and activating JNK1/2 activities attenuates LSCC progression.

This paper contains a "huge" amount of data. Nonetheless, I am unconvinced by the models they utilized to demonstrate the link between LKB1 and JNK in LSCC. More specifically, the authors demonstrate that LKB1 deficiency decreases the level of JNK1/2 phosphorylation (Fig. 2a), as well as MKK7 expression (Fig. 2e) in SCC. It is therefore very unclear why they chose to examine the impact of MKK7 ablation in a cancer cell line that already displays impaired JNK activation, i.e. LSCC cell line generated from *Lkb1^{d/d}Pten^{d/d}* tumor. Likewise, the experiments presented in Fig. 3 and supplementary Figs. 5 and 6, examined the effect of *Jnk1/2* ablation in *Lkb1* deficient LSCC cell line and tumors, which are already defective in JNK signalling. In light of the ability of IL-1 β , TNF α , and anisomycin to activate JNK in the *Lkb1;Pten* deficient LSCC cell line, thereby causing a decrease in Δ Np63/P63 (Fig. 3f), one could even argue that JNK negatively controls markers of LSCC, independently of LKB1. This conclusion does not fit well with the idea that JNK is required for mediating the tumour suppressive function of LKB1. Maybe the authors are looking at two distinct pathways that act in parallel. For example, the tumour suppressive function of JNK in LSCC may be associated with its pro-apoptotic activity as demonstrated in Fig 4a-c.

To convincingly establish a link between LKB1 and JNK signaling in LSCC formation, the authors should restore a normal level of MKK7 expression in *LKB1^{d/d}Pten^{d/d}* cancer cells and demonstrate that this restores JNK phosphorylation and reverts the oncogenic phenotype, i.e. by showing decreased level of SCC markers p63/ Δ Np63. Results of this experiment would easily replace panel g in Figure 2.

It is more difficult to suggest a replacement for Fig. 3. Ideally, this would require a knock in mouse model expressing a constitutive active mutant of MKK7 or a phosphomimetic JNK mutant (i.e. the replacement of Tyr/Thr in the active site by glutamic acid residues).

The authors should revise their paper and clarify their conclusion in light of these comments.

Minor comments:

Supplementary figure 3: The authors state in the text (P7L152) that the *LKB1^{d/d}Pten^{d/d}* mouse model exhibits "an incidence rate of lung adenosquamous cell carcinomas (ASCs) at 88.9%". In panel e, the rate of 88.9 is given for both SCC-DS and SCC. This is confusing and should be clarified. There should also verify the labelling and numbers reported in Fig. 3e.

Figure legend 2c: There is no lower panel, i.e. "WB analysis of protein levels in 1-month-old mouse lungs (n = 3)". Please correct.

P11 Line 270-277: The argument that "activation of JNK1/2 may be an option for LSCC patients with low PD-L1 expressing" is based on very weak evidence (Fig 5F). Unless, the authors can further substantiate this statement, this paragraph should be removed together with Fig. 5F.

Reviewer #2, Expertise: LSCC and mouse models
(Remarks to the Author):

In this manuscript, Liu et al. show that *Lkb1* deletion in CCSP-positive cells of the lung is sufficient for the development of SCC, adenocarcinoma, and mixed lesions (adenosquamous carcinoma). Moreover, concomitant *Jnk1/2* deletion accelerates tumorigenesis, and pharmacologic JNK activation appears to be inhibitory, though this latter point needs additional data to be fully supported. Clinical correlations between a gene signature of JNK activity and survival in patients with SCC are also provided.

Major concerns

1. The analysis of *in vivo* anisomycin administration (Figure 4d-e) is incomplete with respect to both the existing experiment and other key experiments that are needed to support the authors' conclusions.
 - a. Existing data: It is not clear whether Figure 4d represents quantitation of % of mice with macroscopic or microscopic tumors. Regardless, microscopic analysis of these mice is not presented and is essential for understanding the effects of anisomycin *in vivo*. For example, the authors report that *Lkb1/Pten*-null mice develop both SCC and adenocarcinoma (and a variety of precursor lesions). Are SCCs selectively impaired by anisomycin, or are multiple types of neoplasia inhibited? The authors provide high quality microscopic analyses in other figures of the manuscript – this should be done for these experiments as well. The authors should also clarify whether drug was given for 1 month (results section) or 8 weeks (as stated in methods).
 - b. New experiments: The key experiment that is not included here would be to treat *Lkb1/Jnk1/2*-mutant mice with anisomycin and show that these tumors are minimally affected by the drug in comparison with *Jnk1/2*-wild type tumors. Although this is a relatively large experiment, the conclusions of the manuscript would be much stronger with these data in hand. Ideally, the authors would also analyze the effects of short-term treatment of established tumors with anisomycin. This would support their contention that JNK activators might be useful therapeutics in SCC.
2. The authors should describe more precisely whether they think that the SCCs in this model arise *de novo*, from adenocarcinoma, or that both scenarios occur frequently. The first possibility seems to be favored in the discussion based on anatomic location. However, the classification of “adenocarcinoma with squamous differentiation” and AdSCCs as part of the SCC-DS would be more consistent with the second possibility. (If the authors do think that most SCC's arise *de novo*, then I don't think it makes sense to include “adenocarcinoma with squamous differentiation” and AdSCCs as part of the SCC-DS.) This is an important distinction because human SCC and adenosquamous carcinoma are very different diseases that should not be conflated in mouse models. The supplemental tables providing details on neoplasia identified in each mouse are very helpful, but there could be a better summary of the data in the main text. For example, the tables describing the % of mice with each type of lesion can be better described (table 1 and tables within the figures). Are these percentages partially overlapping or do they represent distinct subsets of mice? Overall, it would be helpful to report how many mice have only SCC lesions and non-adenocarcinoma precursors, with no evidence of possible adenocarcinoma precursors. Relevant to this point, what is the frequency of LKB1 mutation in pure human SCC vs. adenosquamous carcinoma?
3. Figure 2g/h. Does exogenous MKK7 restore p-JNK1/2 levels in MKK7-null cells? What is the effect of sgRNA against MKK7 and LKB1 on proliferation and survival in these cell lines?
4. Figure 3f. It looks like anisomycin causes a decrease in p63 levels even in *Jnk1/2*-null cells. How much of the effect of the drug is truly JNK dependent?

Minor concerns

1. The impact of *Lkb1* deletion on p-JNK levels does not appear to be entirely direct (Fig 2b/d).

Histologically normal Lkb1-deficient airways have uniform p-JNK, whereas hyperplastic lesions start to lose p-JNK positivity. Does this imply that proliferation induced by Lkb1 loss occurs prior to the decline in pJNK levels? The authors should explain their interpretation more clearly.

2. A caveat of the CCSP-Cre model is that it should not be active in basal cells of the proximal airway, which are p63-positive and are generally thought to be one of the main cells of origin. This should be pointed out in the discussion and the authors could comment on whether they think the JNK pathway is relevant in SCC cells-of-origin that already express p63.

3. In vitro experiments are performed in a very small number of cell lines.

4. The paragraph on PD-L1 at the end of the results seems like a speculative tangent that doesn't add much to the manuscript.

5. Discussion section: Can the authors comment on how JNK signaling might increase DNp63 expression?

Response letter

Reviewer #1,

Comment 1

This paper contains a “huge” amount of data. Nonetheless, I am unconvinced by the models they utilized to demonstrate the link between LKB1 and JNK in LSCC. More specifically, the authors demonstrate that LKB1 deficiency decreases the level of JNK1/2 phosphorylation (Fig. 2a), as well as MKK7 expression (Fig. 2e) in SCC. It is therefore very unclear why they chose to examine the impact of MKK7 ablation in a cancer cell line that already displays impaired JNK activation, i.e. LSCC cell line generated from *Lkb1d/dPten/d* tumor.

Response 1

Using genetically engineered mouse models, we demonstrated that the loss of *Lkb1* or the combination of *Lkb1* and *Pten* in the pulmonary epithelium causes LSCC with a progressive decrease in JNK1/2 activity. JNK1/2 mRNA did not change in this model. However, by using a bioinformatic approach and a Kinome assay, we demonstrated that p-JNK1/2 was altered. To arrive at a mechanistic link between LKB1 and p-JNK1/2, we demonstrated that the mRNA and protein expression of MKK7 was altered.

We agree with the reviewer that the use of the cell line with the loss of LKB1 and already reduced levels of MKK7 was not the best way of demonstrating the link between *Lkb1* and p-JNK1/2 since the levels of MKK7 would only be further reduced. We have removed the data using this mouse cell line from the manuscript. However, in the initial submission, we also examined the impact of MKK7 ablation in a nontumorigenic human bronchial epithelial cell line NL20, which does not have impaired JNK activation. The result (Figure 2i in the initial version) was similar to the impact of remaining MKK7 in the LSCC cell line generated from a *Lkb1d/dPten/d* tumor. In the revised version, we confirmed the results in another nontumorigenic human bronchial epithelial cell line BEAS-2B (Supplementary Fig. 5e).

In addition, we performed new experiments to demonstrate the link between LKB1 and p-JNK1/2 in LSCC by restoring MKK7 expression in the *Lkb1d/dPten/d* mouse LSCC cell line according to Comment 3. Please find the related data in **Response 3**.

Below is the text (Lines 190-194) that addressed this issue and the supporting data (Fig. 3g and Sup. Fig. 5e).

Lines 190 – 194 in the manuscript: “Meanwhile, at the pre-tumor stage modeled by a nontumorigenic human bronchial epithelial cell line NL20 and Beas-2B, the *MKK7* knockout also had reduced JNK1/2 phosphorylation levels (Fig. 3g and Supplementary Fig. 5e) while *LKB1* ablation decreased levels of both MKK7 expression and phosphorylated JNK1/2 (Fig. 3f and Supplementary Fig. 5d).”

Figure 3g (Figure 2i in the initial version)

Supplementary Fig. 5e

Figure 3g: WB analysis of protein expressions in NL20 cells after knockout of MKK7 or scramble (Control) using lentiviral Cas9/gRNA under 5-hour (hr) treatment of TNF α (20 ng/ml).

Sup. Fig. 5e: WB analysis of protein expressions in Beas-2B cells after knockout of MKK7 or scramble (Control) using lentiviral Cas9/gRNA under 5-hour (hr) treatment of TNF α (20 ng/ml).

Comment 2

Likewise, the experiments presented in Fig. 3 and supplementary Figs. 5 and 6, examined the effect of *Jnk1/2* ablation in *Lkb1* deficient LSCC cell line and tumors, which are already defective in JNK signaling. In light of the ability of IL-1 α , TNF β , and anisomycin to activate JNK in the *Lkb1*;*Pten* deficient LSCC cell line, thereby causing a decrease in Δ Np63/P63 (Fig. 3f), one could even argue that JNK negatively controls markers of LSCC, independently of LKB1. This conclusion does not fit well with the idea that JNK is required for mediating the tumor suppressive function of LKB1. Maybe the authors are looking at two distinct pathways that act in parallel. For example, the tumor suppressive function of JNK in LSSC may be associated with its pro-apoptotic activity as demonstrated in Fig 4a-c.

Response 2

To investigate the possibility that JNK1/2 negatively controls LSCC, independently of LKB1, we analyzed the lungs of 12-15-month-old CCSPiCreJnk1f/fJnk2^{-/-} mice (n = 21). There was no squamous cell carcinoma (SCC) formation (New data: Sup. Figure 6g and Supplementary Table 1). There are only sporadic adenoma/adenocarcinoma (33.3%) with or without inflammation.

In our original submission, we reported that CCSPiCreLkb1f/fJnk1f/fJnk2^{-/-} mice started to develop LSCC tumors around 4-month old (Fig. 4b) while in comparison, CCSPiCreLkb1f/f mice started to develop SCC tumors around 11-month old (Table 1). We conclude that Jnk1/2 ablation accelerates Lkb1-deficiency-induced LSCC progression. These *in vivo* data indicate that LKB1 deficiency background is required for JNK1/2 to regulate SCC development.

Collectively, these data plus our other initial findings conclude that JNK1/2 is downstream of LKB1 in the regulation of lung SCC development. To clarify the main conclusion from this study, we have changed the title to “JNK1/2 represses *Lkb1*-deficiency-induced Lung Squamous Cell Carcinoma Progression”.

Please find the related changes in the text (Lines 223- 226) and the supporting data (Sup. Fig. 6g) below.

Lines 223- 226 in manuscript: “Interestingly, there is no LSCC formation in *Jnk1*^{d/d}*Jnk2*^{-/-} mice, which only has adenoma/adenocarcinoma development (33.3%) (Supplementary Fig. 6g and Supplementary Table 1), concluding that the JNK1/2 pathway regulates LSCC development under a *Lkb1* deficient background.”

New data: Sup. Fig. 6g

Sup. Fig. 6 g: Representative H&E staining of lung adenocarcinoma in 12-15-month-old *Jnk1*^{d/d}*Jnk2*^{-/-} mice (Supplemental Table 1). Scale bar: 500 μ m (Black box, left panel); 50 μ m (Black box, right panel).

Comment 3

To convincingly establish a link between LKB1 and JNK signaling in LCSS formation, the authors should restore a normal level of MKK7 expression in LKB1d/dPtend/d cancer cells and demonstrate that this restores JNK phosphorylation and reverts the oncogenic phenotype, i.e. by showing decreased level of SCC markers p63/ Δ Np63. Results of this experiment would easily replace panel g in Figure 2.

Response 3

Per the suggestion of the reviewer, we performed experiments to demonstrate the link between LKB1 and p-JNK1/2 in LSCC by restoring MKK7 expression in the Lkb1d/dPtend/d mouse LSCC cell line. We observed that P-JNK1/2 was increased while Δ Np63 and p63 were decreased after restoring MKK7 expression in these cells (New data: Figure 3d-e).

In the revised version, we used this MKK7 overexpression result to replace the MKK7 ablation results in the Lkb1d/dPtend/d mouse LSCC cell line. Meanwhile, similar results were observed in human LSCC cells with mutation of LKB1 and PTEN (New data: Sup. Figure 5c).

Please find the related changes in text (Lines 187-190) and supporting data (Figure 3d-e and Sup. Figure 5c) as below.

Lines 187-190 in manuscript: "Exogenous expression of MKK7 in mLSCC by transfection restored JNK1/2 phosphorylation and decreased the levels of p63 and Δ Np63 (Fig. 3d-e). Similar results were observed in H157 cells, which are human LSCC cells with the mutations of LKB1 and PTEN (Supplementary Fig. 5c)."

Figure 3d-e

Figure 3d-e: d, WB analysis of protein expressions in mouse lung SCC cells (mLSCC^{LP.3}). e, Co-staining of Flag/MKK7 and pJNK1/2 or p63 or Δ Np63 in mouse LSCC cells. Scale bar: 50 μ m; Yellow arrows point out the cells expressing the exogenous Flag-MKK7; White arrows point out the cells without clear expression of the exogenous Flag-MKK7.

Sup. Figure 5c

Sup. Figure 5c: Co-staining of Flag/MKK7 and pJNK1/2 or p63 or Δ Np63 in human LSCC cells. Scale bar: 50 μ m; Yellow arrows point out the cells expressing the exogenous Flag-MKK7; White arrows point out the cells without clear expression of the exogenous Flag-MKK7.

Comment 4

It is more difficult to suggest a replacement for Fig. 3. Ideally, this would require a knock in mouse model expressing a constitutive active mutant of MKK7 or a phosphomimetic JNK mutant (i.e. the replacement of Tyr/Thr in the active site by glutamic acid residues).

The authors should revise their paper and clarify their conclusion in light of these comments.

Response 4

We agree that a knock-in mouse model expressing a constitutive active mutant of MKK7 or a phosphomimetic JNK mutant is an ideal genetic model to prove JNK1/2 activity in preventing LCSS development. These valuable suggestions are part of our future work. However, we have attempted *in vitro* experiments to address this issue, as described in **Response 3**.

In addition, we provided new data to show JNK1/2 activity prevents LCSS development by using a JNK1/2 activator. In detail, we further analyzed the microscopic tumors of $Lkb1^{d/d} Pten^{d/d}$ mice, which are treated by Anisomycin (JNK1/2 activator) (New data: Figures 5d-e). We also demonstrate that the $Lkb1/Jnk1/2$ -mutant mouse tumors are minimally affected by Anisomycin (New data: Figure 5f and Supplementary Table 1) in comparison to $Jnk1/2$ -wild type tumors (New data: Figures 5d-e).

Please find the related changes in the text (Lines 263 – 266) and the supporting data (Figures 5d-f) below.

Lines 263 – 266 in manuscript: “Consistent with the *in vitro* results, treatment of $Lkb1^{d/d} Pten^{d/d}$ mice with Anisomycin for 8 weeks significantly inhibited LSCC formation (Fig. 5d-e) and lung tumor development (Supplementary Fig. 7c). In contrast, Anisomycin did not affect the development of LSCC and lung tumors in $Lkb1^{d/d} Jnk1^{d/d} Jnk2^{-/-}$ mice (Fig. 5f and Supplementary Table 1).”

New data: Figures 5d-f

d

e

f

Figure 5d-f: d, Representative H&E staining of lungs or lung tumors of the mice treated with DMSO or Anisomycin. e, Quantification of mice with lung SCC or SCC-DS at the endpoint of treatment (Fig. 5d). The b group of Lkb1^{d/d} Pten^{d/d} had eight mice and the remainder of each group included nine mice. χ^2 test (Two-way): ** $P < 0.01$; *** $P < 0.001$. f, Representative H&E staining of lungs or lung tumors of the mice treated with DMSO ($n = 9$) or Anisomycin ($n = 10$).

Minor comments:

Comment 5

Supplementary figure 3: The authors state in the text (P7L152) that the LKB1d/dPten^{d/d} mouse model exhibits “an incidence rate of lung adenosquamous cell carcinomas (ASCs) at 88.9%”. In panel e, the rate of 88.9 is given for both SCC-DS and SCC. This is confusing and should be clarified. There should also verify the labelling and numbers reported in Fig. 3e.

Response 5

We apologize for the confusion. In Sup. Figure 3e, the rate of 88.9 for both SCC-DS and SCC is correct. Sup. Table 2 includes detailed information. Therefore, we added the citation of Sup. Table 2 in the text (Lines 149-155). Please find the detail information in Sup. Table 2. Below is the correction in the text.

Lines 153-159 in the manuscript: “We took advantage of the CCSP^{Cre}Lkb1^{fl/fl}Pten^{fl/fl} (Lkb1^{d/d}Pten^{d/d}) mouse model that has accelerated LSCC development in 3 months (Supplementary Fig. 3e-h; Supplementary Tables 1-2 and 4), an incidence rate of lung adenosquamous cell carcinomas (ASCs) at 88.9% (Supplementary Fig.

3e and Supplementary Table 2), nearly identical molecular profiles between lung SCC and ASC (Supplementary Fig. 3f-i), close resemblance of the combined SCC/ASC transcriptome profile to that of human lung SCCs (Supplementary Fig. 3j and Supplemental Table 3).”

Comment 6

Figure legend 2c: There is no lower panel, i.e. “WB analysis of protein levels in 1-month-old mouse lungs (n = 3)”. Please correct.

Response 6

We have corrected this in the manuscript (Lines 770-774) and the revised figure legend was listed as below.

Lines 770-774 in the manuscript (original Figure 2c has been organized into Figure 2e): “Kinome array and Western Blot analyses. Kinome array was performed in 1-month-old (1M) mouse lungs (n = 6) (see Supplementary Fig. 4c-d for a full summary of kinome array). The dots of anti-phosphorylated (P)-proteins in the membranes; The table listing the top changed P-proteins in 1M *Lkb1^{d/d}Pten^{d/d}* lungs (pre-tumor stage) compared to 1M wild type (WT) lungs.”

Comment 7

P11 Line 270-277: The argument that “activation of JNK1/2 may be an option for LSCC patients with low PD-L1 expressing” is based on very weak evidence (Fig 5F). Unless, the authors can further substantiate this statement, this paragraph should be removed together with Fig. 5F.

Response 7

As reviewer suggested, we have removed the whole paragraph on PD-L1 and the related data.

Reviewer #2,

Major concerns

Comment 8

1. The analysis of in vivo anisomycin administration (Figure 4d-e) is incomplete with respect to both the existing experiment and other key experiments that are needed to support the authors’ conclusions.

a. Existing data: It is not clear whether Figure 4d represents quantitation of % of mice with macroscopic or microscopic tumors. Regardless, microscopic analysis of these mice is not presented and is essential for understanding the effects of anisomycin in vivo. For example, the authors report that *Lkb1/Pten*-null mice develop both SCC and adenocarcinoma (and a variety of precursor lesions). Are SCCs selectively impaired by anisomycin, or are multiple types of neoplasia inhibited? The authors provide high quality microscopic analyses in other figures of the manuscript – this should be done for these experiments as well. The authors should also clarify whether drug was given for 1 month (results section) or 8 weeks (as stated in methods).

Response 8

In the initial submission version, Figure 4d-e represents quantitation of % of mice with macroscopic tumors. Anisomycin was given for 8 weeks (as stated in methods), not for 1 month. We have made the changes accordingly.

Per the suggestion of Reviewer 2, we reanalyzed these Anisomycin-treated *Lkb1/Pten*-null mice by H&E staining. Similar pathological evaluation reports were independently provided by pathologists. Here we observed the significant inhibition of lung SCC development (New data: Figures 5d-e and Revised Sup. Table 1).

Meanwhile, the effect of Anisomycin on adenoma/adenocarcinoma development was not observed. The percentage of adenoma/adenocarcinoma (22.2 - 33.3%) in these *Lkb1/Pten*-null mice was described in the initial submission (Sup. Figure 3e and Sup. Table. 4). The percentage of adenoma/adenocarcinoma was also low in *Lkb1/Pten*-null mouse group treated with DMSO (22.2-25%) or with Anisomycin (11.15 - 25%) (New data: Revised Sup. Table 1).

Please find the revised paragraph (Lines 263-265) and new data (Figure 5d-e) in the revised manuscript.

Lines 263-265 in the manuscript: “Consistent with the *in vitro* results, treatment of *Lkb1^{d/d}Pten^{d/d}* mice with Anisomycin for 8 weeks significantly inhibited LSCC formation (Fig. 5d-e) and lung tumor development (Supplementary Fig. 7c), with no obvious inhibition on lung adenoma/adenocarcinoma (Sup. Table 1).”

New data: Figures 5d-e

d

e

Figure 5d-e: d, Representative H&E staining of lungs or lung tumors of the mice treated with DMSO or Anisomycin. e, Quantification of mice with lung SCC or SCC-DS at the endpoint of treatment (Fig. 5d). The b group of *Lkb1^{d/d}Pten^{d/d}* had eight mice and the remainder of each group included nine mice. χ^2 test (Two-way): ** $P < 0.01$; *** $P < 0.001$.

Comment 9

b. New experiments: The key experiment that is not included here would be to treat *Lkb1/Jnk1/2*-mutant mice with anisomycin and show that these tumors are minimally affected by the drug in comparison with *Jnk1/2*-wild type tumors. Although this is a relatively large experiment, the conclusions of the manuscript would be much stronger with these data in hand. Ideally, the authors would also analyze the effects of short-term treatment of established tumors with anisomycin. This would support their contention that JNK activators might be useful therapeutics in SCC.

Response 9

Per the suggestion of the reviewer, *Lkb1/Jnk1/2*-mutant mice were treated for 8 weeks using DMSO and Anisomycin (2.5mg/kg). In contrast to that of *Jnk1/2*-wild type tumors of *Lkb1/Pten*-null mice (New data: Figures 5d-e and Revised Sup. Table 1), Anisomycin treatment did not inhibit *Jnk1/2*-deficient SCC development in *Lkb1/Jnk1/2*-mutant mice (New data: Figures 5f and Revised Sup. Table 1). These data suggest that wild type *Jnk1/2* was required for Anisomycin to inhibit lung SCC development.

Please find the revised paragraph (Lines 263-267) and New data (Figures 5d-f) in the revised manuscript.

With respect to treating established tumors, this would require *in vivo* imaging to identify when tumors are present. This would require a reporter mouse line has to be introduced into our current tumor mouse models. Given the revision time length this was not feasible. We considered it to be our future experiments.

Lines 263-267 in the manuscript: “Consistent with the *in vitro* results, treatment of *Lkb1^{d/d}Pten^{d/d}* mice with Anisomycin for 8 weeks significantly inhibited LSCC formation (Fig. 5d-e) and lung tumor development (Supplementary Fig. 7c), with no obvious inhibition on lung adenoma/adenocarcinoma (Sup. Table 1). In contrast, Anisomycin did not affect the development of LSCC and lung tumors in *Lkb1^{d/d}Jnk1^{d/d}Jnk2^{-/-}* mice (Fig. 5f and Supplementary Table 1).”

New data: Figures 5d-f

Figure 5d-f: d, Representative H&E staining of lungs or lung tumors of the mice treated with DMSO or Anisomycin. e, Quantification of mice with lung SCC or SCC-DS at the endpoint of treatment (Fig. 5d). The b group of *Lkb1^{d/d}Pten^{d/d}* had eight mice and the remainder of each group included nine mice. χ^2 test (Two-way): ** $P < 0.01$; *** $P < 0.001$. f, Representative H&E staining of lungs or lung tumors of the mice treated with DMSO ($n = 9$) or Anisomycin ($n = 10$).

Comment 10

2. The authors should describe more precisely whether they think that the SCCs in this model arise *de novo*, from adenocarcinoma, or that both scenarios occur frequently. The first possibility seems to be favored in the discussion based on anatomic location. However, the classification of “adenocarcinoma with squamous differentiation” and AdSCCs as part of the SCC-DS would be more consistent with the second possibility. (If the authors do think that most SCC’s arise *de novo*, then I don’t think it makes sense to include “adenocarcinoma with squamous differentiation” and AdSCCs as part of the SCC-DS.) This is an important distinction because human SCC and adenosquamous carcinoma are very different diseases that should not be conflated in mouse models.

Response 10

Based on our data and the reports in the related references (PMID: 24531128, PMID: 25115923 and PMID: 25936644), we propose that a portion of the SCCs arises *de novo* and a portion may arise from the transdifferentiation from adenoma or adenocarcinoma.

More specifically, as reviewer’s points, SCCs in our mouse model arise *de novo* based on anatomic location (Figures 1b and 4b, Sup. Figure 1c-e and Sup. Table 2). Our CCSP^{iCre} has been showed to have Cre activity in the basal cells of the proximal airway (PMID: 25753424), which are p63-positive and are generally thought to be one of the main cells of origin (PMID: 27728803). This suggest that some of SCC cells were developed from these p63 positive basal cells. The observation of SCC only in Lkb1d/d, Lkb1d/dPten^{d/d} and Lkb1d/dJnk1d/dJnk2^{-/-} also supports this idea (Revised data: Sup. Tables 1-2). Meanwhile, a second developing cohort of SCCs was indicated by the SCC marker staining (CK5 and P63), which was positive in SCC-DSs of these models (Sup. Figure 1f-g and 3g and 6d-f, and Figure 4c-d, and Sup. Table 2), especially for these adenocarcinoma precursors. Considering the reports of the trans-differentiation from Lkb1-deficient adenocarcinoma (AD) into SCC in Ad-Cre-Kras^{G12D}Lkb1^{ff} mice (PMID: 24531128, PMID: 25115923 and PMID: 25936644), the second possibility also exists.

In the future, the use of single cell RNA-Seq on the lungs of these models at varying stages of development will help identify the changes in cell types and the transcriptome of these cells during SCC progression. This will aid in the understanding the cell origin of SCC.

Here we added the following paragraph (Lines 432-448) to discuss the cell origins of SCCs in our SCC models.

Lines 432-448 in the manuscript: “The LSCC model resulting from Lkb1 loss in the pulmonary epithelium represents a model to study the cell of origins of SCCs. Anatomic location precursors suggest that SCCs in our mouse model arise *de novo* (Figure 1b and 4b, Sup. Figure 1c-e and Sup. Table 2). Our CCSP^{iCre} has been shown to have Cre activity in the basal cells of the proximal airway¹³, which are p63-positive and are generally thought to be one of the main cell of origins¹¹. This suggests that some of SCC cells were developed from these p63 positive basal cells. The observation of SCC only in Lkb1^{d/d} (8.9%), Lkb1^{d/d}Pten^{d/d} (11.1%) and Lkb1^{d/d}Jnk1^{d/d}Jnk2^{-/-} (16.7%) mice also supports this idea (Sup. Tables 1-2). Here, we defined the mice of SCC only as the ones that have only SCC lesions and non-adenocarcinoma precursors. Meanwhile, SCCs may also transdifferentiate from adenocarcinoma or its precursors. This idea was indicated by the SCC marker staining (CK5 and P63), which was positive in SCC-DSs of these models (Sup. Figure 1f-g and 3g and 6e-f, and Figure 4c-d, and Sup. Table 2), especially for these adenocarcinoma precursors. Considering the reports of the trans-differentiation from Lkb1-deficient adenocarcinoma (AD) into SCC in Ad-Cre-Kras^{G12D}Lkb1^{ff} mice^{37,61,62}, this possibility also exists. The use of single cell RNA-Seq tumors for this model at varying stages of development will help identify the changes in cell types and the transcriptome of these cells during SCC progression. This will aid in the understanding the cell of origins of SCC.”

Comment 11

The supplemental tables providing details on neoplasia identified in each mouse are very helpful, but there could be a better summary of the data in the main text. For example, the tables describing the %

of mice with each type of lesion can be better described (table 1 and tables within the figures). Are these percentages partially overlapping or do they represent distinct subsets of mice? Overall, it would be helpful to report how many mice have only SCC lesions and non-adenocarcinoma precursors, with no evidence of possible adenocarcinoma precursors.

Response 11

We added more details on neoplasia in the main text (Lines 104-109, 153-161, 208-226 and 437-439). These subtype percentages are partially overlapped. Some mice only have SCC lesions, independently confirmed by two pathologists. In detail, SCC lesions were only observed in *Lkb1*^{d/d} (8.9%), *Lkb1*^{d/d}*Pten*^{d/d} (11.1%) and *Lkb1*^{d/d}*Jnk1*^{d/d}*Jnk2*^{-/-} (16.7%) (Sup. Tables 1-2).

Lines 104-109 in the manuscript: “Meanwhile, progressive lung SCC developmental stages (SCC-DSs) were also observed after *Lkb1* ablation (Supplementary Tables 1-2), including epithelial hyperplasia (5.4%) (Supplementary Fig. 1e), squamous metaplasia (1.8%) (Supplementary Fig. 1e), adenocarcinoma with squamous differentiation (10.7%) (Supplementary Fig. 1f) and adenosquamous carcinoma (ASC) (5.4%) (Supplementary Fig. 1g).”

Lines 153-161 in the manuscript: “We took advantage of the *CCSP*^{iCre}*Lkb1*^{ff/ff}*Pten*^{ff/ff} (*Lkb1*^{d/d}*Pten*^{d/d}) mouse model that has accelerated LSCC development in 3 months (Supplementary Fig. 3e-h; Supplementary Tables 1-2 and 4), an incidence rate of lung adenosquamous cell carcinomas (ASCs) at 88.9% (Supplementary Fig. 3e and Supplementary Table 2), nearly identical molecular profiles between lung SCC and ASC (Supplementary Fig. 3f-i), close resemblance of the combined SCC/ASC transcriptome profile to that of human lung SCCs (Supplementary Fig. 3j and Supplemental Table 3). In addition, epithelial hyperplasia (11.1%) and epithelial hyperplasia with squamous metaplasia (11.1%) were observed in the *Lkb1*^{d/d}*Pten*^{d/d} mouse model (Supplementary Tables 1-2).”

Lines line 208-226 in the manuscript: “*Lkb1*^{d/d}*Jnk1*^{d/d}*Jnk2*^{-/-} mice started to develop squamous metaplasia in the lungs at 4 months old, followed by full penetrance of pathologic phenotypes (CK5+, P63+) at 7 months old and beyond (Fig. 4b and Supplementary Fig. 6d). The associated phenotypes include epithelial hyperplasia (66.7%) (Fig. 4c), epithelial hyperplasia with squamous metaplasia (25%) (Fig. 4b), and adenocarcinoma with squamous differentiation (16.7%) (Supplementary Fig. 6e), adenosquamous carcinoma (25%) (Fig. 4d) and SCC (33.3%) (Supplementary Fig. 6d) (Supplementary Tables 1-2). Compared to 11-14-month-old *Lkb1*^{d/d} mice that showed epithelial hyperplasia (5.4%) (Supplementary Fig. 1c-d), epithelial hyperplasia with squamous metaplasia (1.8%) (Supplementary Fig. 1e), squamous metaplasia (1.8%), adenocarcinoma with squamous differentiation (10.7%) (Supplementary Fig. 1f), adenosquamous carcinoma (5.4%) (Supplementary Fig. 1g), SCC (16.1%) (Fig. 1a-b) (Supplementary Tables 1-2), and loss of *Jnk1/2* in the *Lkb1* deficient background accelerated LSCC development in vivo. *Lkb1*^{d/d}*Jnk1*^{d/d}*Jnk2*^{-/-} mice also developed adenocarcinoma (Supplementary Fig. 6f). Although the ADs generally showed a papillary morphology microscopically, they resembled LSCCs in that they are positive for IHC expression of the SCC markers p63 and CK5 (Supplementary Fig. 6f). Interestingly, there is no LSCC formation in *Jnk1*^{d/d}*Jnk2*^{-/-} mice, which only has adenoma/adenocarcinoma development (33.3%) (Supplementary Fig. 6g and Supplementary Table 1), concluding that the JNK1/2 pathway regulates LSCC development under a *Lkb1* deficient background.”

Lines 437-439 in the manuscript: “The observation of SCC only in *Lkb1*^{d/d} (8.9%), *Lkb1*^{d/d}*Pten*^{d/d} (11.1%) and *Lkb1*^{d/d}*Jnk1*^{d/d}*Jnk2*^{-/-} (16.7%) mice also supports this idea (Sup. Tables 1-2).”

Comment 12

Relevant to this point, what is the frequency of LKB1 mutation in pure human SCC vs. adenosquamous carcinoma?

Response 12

Following the suggestion of the reviewer, we analyzed the frequency of LKB1 mutation in pure human SCC vs. adenosquamous carcinoma (ASC) in human lung cancer using the COSMIC database (<https://cancer.sanger.ac.uk/cosmic>), which includes 1,588 SCC and 36 ASC samples among 8,018 patients. Based on the information of “AA Mutation” or “CDS Mutation”, we found 30 and 4 mutations in SCC and ASC samples, respectively. Therefore, the frequency of LKB1 mutation in pure human SCC is 1.89% and 11.11% in ASC. In another word, the mutation rate in ASC is much higher (5.89 fold) than SCC. Meanwhile, we observed one common mutation (CDS: c.1062C>G; AA: p.F354L) between SCC and ASC. These results suggest the possibility of trans-differentiation from ASC into SCC.

Here we added the following paragraph (Lines 449-456) to discuss the frequency of LKB1 mutation in human.

Lines 449-456 in the manuscript: “Interestingly, the COSMIC database (<https://cancer.sanger.ac.uk/cosmic>), which includes 1,588 SCC and 36 ASC samples among 8,018 patients, shows that the frequency of LKB1 mutations in pure human SCC and ASC is 1.89% and 11.11%, respectively. In other words, the mutation rate in ASC is much higher (5.89 fold) than SCC. Therefore, LKB1 may require secondary mutations, such as PTEN or alterations in JNK1/2 signaling to direct the progression of the de novo tumors to SCC vs ASC or to promote the progression of ASC to SCC. Meanwhile, we observed one common mutation (CDS: c.1062C>G; AA: p.F354L) between SCC and ASC, which may indicate the potential for transdifferentiating.”

Comment 13

3. Figure 2g/h. Does exogenous MKK7 restore p-JNK1/2 levels in MKK7-null cells?

Response 13

Per the suggestion of the reviewer, we performed experiments to restore MKK7 expression in the Lkb1d/dPtend/d mouse LSCC cell line. We observed that P-JNK1/2 was increased while Δ Np63 and p63 were decreased after restoring MKK7 expression in these cells (New data: Figure 3d-e).

In the revised version, we used this MKK7 overexpression result to replace the MKK7 ablation results in the Lkb1d/dPtend/d mouse LSCC cell line. Meanwhile, similar results were observed in human LSCC cells with mutation of LKB1 and PTEN (New data: Sup. Figure 5c).

Please find the related changes in text (Lines 187-190) and supporting data (Figure 3d-e and Sup. Figure 5c) as below.

Lines 187-190 in manuscript: “Exogenous expression of MKK7 in mLSCC by transfection restored JNK1/2 phosphorylation and decreased the levels of p63 and Δ Np63 (Fig. 3d-e). Similar results were observed in H157 cells, which are human LSCC cells with the mutations of LKB1 and PTEN (Supplementary Fig. 5c).”

Figure 3d-e

Figure 3d-e: d, WB analysis of protein expressions in mouse lung SCC cells (mLSCC^{LP.3}). e, Co-staining of Flag/MKK7 and pJNK1/2 or p63 or ΔNp63 in mouse LSCC cells. Scale bar: 50μm; Yellow arrows point out the cells expressing the exogenous Flag-MKK7; White arrows point out the cells without clear expression of the exogenous Flag-MKK7.

Sup. Figure 5c

Sup. Figure 5c: Co-staining of Flag/MKK7 and pJNK1/2 or p63 or ΔNp63 in human LSCC cells. Scale bar: 50μm; Yellow arrows point out the cells expressing the exogenous Flag-MKK7; White arrows point out the cells without clear expression of the exogenous Flag-MKK7.

Comment 14

What is the effect of sgRNA against MKK7 and LKB1 on proliferation and survival in these cell lines?

Response 14

Knockout of MKK7 and LKB1 in human bronchial epithelial cells increases cell growth (New data: Figure 3h and Sup. Figure 5d-f).

Please find the related changes in text (Lines 195-196) and supporting data below.

Lines 195-196 in the manuscript: "Furthermore, ablation of LKB1 and MKK7 in NL20 and Beas-2b promoted cell growth (Fig. 3h and Supplementary Fig. 5f)."

New data: Figure 3h and Sup. Figure 5f

Figure 3h

Sup. Figure 5f

Figure 3h: MTS assay analysis of cell viability of NL20. -: parental cells; gControl: gRNA targeting non-coding region; gLKB1: gRNA targeting LKB1; gMKK7: gRNA targeting MKK7.

Figure 5f: MTS assay analysis of cell viability of Beas-2B. -: parental cells; gControl: gRNA targeting non-coding region; gLKB1: gRNA targeting LKB1; gMKK7: gRNA targeting MKK7.

Comment 15

4. Figure 3f. It looks like anisomycin causes a decrease in p63 levels even in Jnk1/2-null cells. How much of the effect of the drug is truly JNK dependent?

Response 15

The truly JNK dependent effect of Anisomycin-induced decrease of p63 and Δ Np63 is hard to calculate since we used the pool cells with the knockout of *Jnk1/2*. Here, we did the semi-quantification of the bands of P63 and Δ Np63 in Figure 3f to compare the changes affected by Anisomycin in gRNA-control and gRNA-Jnk1/2 groups. Approximately, 46.3% of the decrease of Δ Np63 and 29.1% of the reduce of P63 induced by

Anisomycin is JNK1/2 dependent, suggesting that Anisomycin also can reduce the expression of p63 and Δ Np63 partially independent of JNK1/2.

Please find the related data and changes the manuscript (Lines 238-240).

Lines 243-245 in the manuscript: “Quantification of the protein bands of Δ Np63 and p63 showed JNK1/2 had a stronger effect in repressing Δ Np63 than inhibiting p63 expression (Fig. 4f and Supplementary Fig. 7a).”

Revised Figure 4f

New Data: Sup. Figure 7a

The ratio of bands between Anisomycin treatment and Anisomycin non-treatment			The percent of Anisomycin-induced decrease of Δ Np63 or p63 depending on JNK1/2	
	gControl	gJnk1/2		(gJnk1/2 - gControl) / gJnk1/2 * 100
Δ Np63	0.27 / 1.1 = 0.245	0.73 / 1.6 = 0.456	Δ Np63	(0.456 - 0.245) / 0.456 = 46.3%
p63	0.48 / 1.1 = 0.436	0.8 / 1.3 = 0.615	p63	(0.615 - 0.436) / 0.615 = 29.1%

Figure 4f: WB analysis of protein expressions in mLSCCLP3 cells under 5-hour (hr) treatment of IL-1 β (2 ng/ml), TNF α (20 ng/ml) and Anisomycin (10 μ M). Free medium and culture medium are described in the Method section.

Sup. Figure 7a: Calculation of the bands of p63 and Δ Np63 in Fig. 4f.

Minor concerns

Comment 16

1. The impact of Lkb1 deletion on p-JNK levels does not appear to be entirely direct (Fig 2b/d). Histologically normal Lkb1-deficient airways have uniform p-JNK, whereas hyperplastic lesions start to lose p-JNK positivity. Does this imply that proliferation induced by Lkb1 loss occurs prior to the decline in pJNK levels? The authors should explain their interpretation more clearly.

Response 16

Our current data cannot define the earlier event between proliferation and declined P-JNK1/2 in hyperplastic lesions. It is possible that that proliferation induced by *Lkb1* loss occurs prior to the decline in p-JNK1/2 levels since we observed that knockout of *LKB1* promoted cell proliferation in human bronchial epithelial cells (NL20 and Beas-2B) (New data listed in Response 14: Figure 3h and Sup. Figure 5f)

Meanwhile, based on our findings that JNK1/2 prevents SCC development, the declined P-JNK1/2 may contribute the development of hyperplastic lesions. As a feedback regulation mechanism, these hyperplastic lesions may facilitate the further decline of P-JNK1/2 by providing an oncogenic environment.

Please find the related changes (Lines 169-176 and 309-310) in the manuscript.

Lines 169-176 in the manuscript: "Histologically normal Lkb1-deficient airways have uniform p-JNK1/2, whereas hyperplastic lesions start to lose p-JNK1/2 positivity (Figures 2b and 2f). This suggests that the impact of Lkb1 deletion on p-JNK1/2 levels does not appear to be entirely direct, implying that proliferation induced by Lkb1 loss occurs prior to the decline in p-JNK1/2 levels. The declined P-JNK1/2 levels may contribute to the development of hyperplastic lesions. As a feedback regulation mechanism, these hyperplastic lesions may facilitate the further decline of P-JNK1/2 by providing an oncogenic environment."

Lines 309-310 in the manuscript: "Loss of Lkb1 however does not automatically cause loss of p-JNK1/2 in normal mouse airway epithelial cells. This would indicate that other events must occur to alter p-JNK1/2 levels."

Comment 17

2. A caveat of the CCSP-Cre model is that it should not be active in basal cells of the proximal airway, which are p63-positive and are generally thought to be one of the main cells of origin. This should be pointed out in the discussion and the authors could comment on whether they think the JNK pathway is relevant in SCC cells-of-origin that already express p63.

Response 17

Our previous characterization of CCSP-iCre showed that iCre also had Cre activity in basal cells expressing p63 (Figure S2E, PMID: 25753424). Therefore, we think that the JNK pathway is relevant to SCC cells-of-origin that already express p63, at least in the development of some of SCC cells.

We list this result (Figure S2E, PMID: 25753424) and the related paragraph (Lines 434-439) as below.

*Lines 434-439 in the manuscript: "Our CCSP^{iCre} has been shown to have Cre activity in the basal cells of the proximal airway¹³, which are p63-positive and are generally thought to be one of the main cell of origins¹¹. This suggests that some of SCC cells were developed from these p63 positive basal cells. The observation of SCC only in *Lkb1*^{d/d} (8.9%), *Lkb1*^{d/d}*Pten*^{d/d} (11.1%) and *Lkb1*^{d/d}*Jnk1*^{d/d}*Jnk2*^{-/-} (16.7%) mice also supports this idea (Sup. Tables 1-2)."*

Figure S2E (PMID: 25753424)

Reference (PMID: 25753424): Figure S2E: (E) Immunofluorescence of p63 and β -Gal (β -galactosidase) in CCSPiCre/R26R reporter mouse lungs. The white arrows indicate the cells coexpressing p63 and β -Gal in upper trachea and bronchia. Boxed areas are magnified in the panels directly below.

Comment 18

3. In vitro experiments are performed in a very small number of cell lines.

Response 18

We added H157 (New data: Sup. Figure 5c) and Beas-2B (New data: Sup. Figure 5d-f) cells to confirm the results of mouse lung SCC and NL20 cells, respectively. H157 is a human LSCC cell line with the mutations of LKB1 and PTEN while Beas-2B is an immortalized human bronchial epithelial cell line.

Please find the related changes in the manuscript (Lines 188-197) and the data below.

Lines 188-197 in the manuscript: “Similar results were observed in H157 cells, which are human LSCC cells with the mutations of LKB1 and PTEN (Supplementary Fig. 5c). Meanwhile, at the pre-tumor stage modeled by a nontumorigenic human bronchial epithelial cell line NL20 and Beas-2B, the MKK7 knockout also had reduced JNK1/2 phosphorylation levels (Fig. 3g and Supplementary Fig. 5e) while LKB1 ablation decreased levels of both MKK7 expression and phosphorylated JNK1/2 (Fig. 3f and Supplementary Fig. 5d). These results suggest that decreased MKK7 expression has a negative impact on JNK1/2 phosphorylation levels during LSCC development. Furthermore, ablation of LKB1 and MKK7 in NL20 and Beas-2b promoted cell growth (Fig. 3h and Supplementary Fig. 5f).”

New data: Sup. Figures 5c-f

Sup. Figures 5c-f: c, Co-staining of Flag/MKK7 and pJNK1/2 or p63 or Δ Np63 in human LSCC cells. Scale bar: 50 μ m; Yellow arrows point out the cells expressing the exogenous Flag-MKK7; White arrows point out the cells without clear expression of the exogenous Flag-MKK7. d, WB analysis of protein expressions in Beas-2B cells, an immortalized human bronchial epithelial cell line, after knockout of LKB1 or scramble (Control) using lentiviral Cas9/gRNA with the treatment of TNF α (20 ng/ml). e, WB analysis of protein expressions in Beas-2B cells after knockout of MKK7 or scramble (Control) using lentiviral Cas9/gRNA under 5-hour (hr) treatment of TNF α (20 ng/ml). f, MTS assay analysis of cell viability of Beas-2B. -: parental cells; gControl: gRNA targeting non-coding region; gLKB1: gRNA targeting LKB1; gMKK7: gRNA targeting MKK7.

Comment 19

4. The paragraph on PD-L1 at the end of the results seems like a speculative tangent that doesn't add much to the manuscript.

Response 19

We have removed this paragraph accordingly.

Comment 20

5. Discussion section: Can the authors comment on how JNK signaling might increase DNp63 expression?

Response 20

One of potential links between JNK1/2 and Δ Np63 is c-Jun. We added one paragraph in the Discussion section to describe it.

And we listed this paragraph as below.

Lines 422-431 in the manuscript: *"Investigation of how JNK1/2 signaling inhibits Δ Np63 expression helps further dissect the mechanism of JNK1/2 repressing LSCC development. One potential link between JNK1/2 and Δ Np63 is c-Jun. Δ Np63 has been reported to be negatively regulated by c-Jun in response to Amyloid- β -Induced cell stress⁶⁰ while JNK1/2 is a typical activator of c-Jun³⁴. Interestingly, our microarray analysis of SCCs in *Lkb1^{d/d}* and *Lkb1^{d/d}Pten^{d/d}* mice showed that c-Jun expression is significantly decreased (Sup. Table 3). We also observed the significant reduction of c-Jun in a *Jnk1/2* knockout array (gRNA-*Jnk1/2* vs gRNA-Control) (Sup. Table 3). These results suggest inactivation of the JNK1/2 pathway induces Δ Np63 expression partially due to the decrease of c-Jun. Continued identification of the pathways regulated by JNK1/2 will serve to better target and customize therapy for SCC."*

REVIEWERS' COMMENTS:

Reviewer #1 (Remarks to the Author):

The authors have satisfactorily addressed all my comments.

Reviewer #2 (Remarks to the Author):

I am satisfied with the authors' response to my review.